# Electron transfer rules of minerals under pressure informed by machine learning

Yanzhang Li[1,2,6], Hongyu Wang[3,6], Yan Li [1,2,6] ✉, Huan Ye[1,2], Yanan Zhang[3], Rongzhang Yin [1,2], Haoning Jia[1,2], Bingxu Hou [1,2], Changqiu Wang[1,2], Hongrui Ding[1,2], Xiangzhi Bai [3,4,5] ✉ & Anhuai Lu [1,2] ✉

Electron transfer is the most elementary process in nature, but the existing electron transfer rules are seldom applied to high-pressure situations, such as in the deep Earth. Here we show a deep learning model to obtain the electronegativity of 96 elements under arbitrary pressure, and a regressed unified formula to quantify its relationship with pressure and electronic configuration. The relative work function of minerals is further predicted by electronegativity, presenting a decreasing trend with pressure because of pressure-induced electron delocalization. Using the work function as the case study of electronegativity, it reveals that the driving force behind directional electron transfer results from the enlarged work function difference between compounds with pressure. This well explains the deep high-conductivity anomalies, and helps discover the redox reactivity between widespread Fe(II)-bearing minerals and water during ongoing subduction. Our results give an insight into the fundamental physicochemical properties of elements and their compounds under pressure.

Electron transfer is the most elementary process in nature, which has been playing essential roles in energy transduction, elemental cycling and life activities[1–3]. Of particular interest is the behavior of electrons with respect to extremes such as high pressure, leading to novel phenomena of insulator-metal transitions, superconductivity, highly reactive atoms and abnormal physiochemical properties of condensed matters[4,5]. Numerous well-designed experimental, computational and theoretical studies in chemistry, physics, material sciences and geosciences have provided insights into the unusual phenomena associated with pressure-induced electronic behavior[5–7]. However, the general rules that drive electron transfer at high pressure remain to be revealed.

Electronegativity has been employed as a fundamental quantify to successfully assess the tendency of an atom to attract electrons[8–10]. In parallel, work function is defined as the energy barrier required to move a free electron from the Fermi level of the solids to the infinity, and is used as an important indicator of electron binding energy[11,12]. In planetary sciences, the mineral work function has been applied to interprete the electrostatic migration of charging lunar dust, the contact electrification phenomenon during electrical beneficiation and the energy threshold of emitting photoelectrons[13–15]. The electronegativity of atoms and work function of minerals could also be applied to quantitatively evaluate the electron transfer tendency and directions under high pressures such as explaining some intricate redox interactions among minerals, fluids, melts and volatiles in Earth's deep region up to ~350 gigapascals (GPa)[6,16]. Nevertheless, both the electronegativity and work function may be greatly changed when compressed, ascribed to the pressure-induced changes in electronic states[4,5].

Currently, it is almost impossible to measure the electronegativity and work function under extreme conditions via

[1]Beijing Key Laboratory of Mineral Environmental Function, School of Earth and Space Sciences, Peking University, 100871 Beijing, China. [2]Key Laboratory of Orogenic Belts and Crustal Evolution, School of Earth and Space Sciences, Peking University, 100871 Beijing, China. [3]Image Processing Center, Beihang University, 102206 Beijing, China. [4]State Key Laboratory of Virtual Reality Technology and Systems, Beihang University, 100191 Beijing, China. [5]Advanced Innovation Center for Biomedical Engineering, Beihang University, 100083 Beijing, China. [6]These authors contributed equally: Yanzhang Li, Hongyu Wang, Yan Li. ✉e-mail: liyan-pku@pku.edu.cn; jackybxz@buaa.edu.cn; ahlu@pku.edu.cn

conventional techniques like photoelectron and thermionic emission spectroscopy. The high-throughput computing can try to solve this problem[17,18], but still seems weak in dealing with natural minerals that possess more complex composition and structural characteristics than simple compounds. For example, the ubiquitous existence of isomorphic substituting impurities in minerals makes us have to cope with a dozen elements and thousands of electrons at each pressure. In this situation, the computational burden is extremely heavy because of the exponential increase in the computation with the increasing number of electrons and dimensions (known as curse of dimensionality)[19,20]. The resulting high-dimensional Schrödinger equations are almost unsolvable[20]. Therefore, an efficient data processing strategy is urgently needed to obtain the electronegativity of element under arbitrary pressures and the work function of materials composed of these elements. Only in this way can we reveal the general rules behind electron transfer at high-pressure conditions.

Here, we show an efficient and instant strategy informed by deep learning, to obtain the electronegativity of elements from H to Cm and the relative work function of compounds composed of these elements under arbitrary pressures up to 500 GPa. Based on this, a unified formula describing the changes of electronegativity on pressure and electronic configuration was obtained by a symbolic regression algorithm. It unlocks the quantitative dependence of relative work functions on mineral compositions under continuously varying pressures. By using the pressure-modulated relative work function as the case study of applying electronegativity to geoscience, the phenomena of generally increasing conductivity of minerals from the Earth's crust to the core and high-conductivity anomaly at the deep mineral discontinuity can be well interpreted. In particular, we reveal the electron transfer rules in deep Earth driven by the enlarged interfacial work function difference between two phases with increasing pressure. This enables us to make new predictions on the redox reactivity order of Fe(II)-bearing minerals with $H_2O$ under high pressure, and discover the

great pressure-driven $H_2$ production potential of silicate minerals during ongoing subduction.

## Electronegativity prediction under pressure via deep learning

The electronegativity dataset of this work came from density function theory (DFT) calculations, including 96 elements (from H to Cm) and under 4 different pressures (0, 50, 200, and 500 GPa)[17]. The newly calculated electronegativity from Dong and his co-workers appropriately modified the definition of Mulliken electronegativity for better use at high pressures, and it used the enthalpy as the relevant thermodynamic potential within the framework of homogeneous electron gas model. At zero pressure, Dong's electronegativity has the same energy unit as Mulliken electronegativity (eV) and also has a good linear correlation with Pauling electronegativity (correlation coefficient $R = 0.91$). For example, the electronegativity of rubidium (Rb), zinc (Zn), nitrogen (N) in Pauling scale is 0.8, 1.6, and 3.0, respectively; while that is −0.8, 1.0, and 4.4 eV, respectively, in the Dong's definition. Although Dong's electronegativity dataset has different reference systems and specific data from the other two well-known electronegativity datasets, it successfully extends the applicability of evaluating electron transfer trends between elements to high pressure cases. The predicted stability of $Mg_xFe_y$ compounds at pressures above 100 GPa accords well with the experimental observations[17], indicating Dong's electronegativity after modifying Mulliken electronegativity is specifically suitable for high pressure cases.

Inspired by this useful dataset with limited 384 data points, the deep learning model was used to obtain the electronegativity of these 96 elements under continuously varying pressures (0–500 GPa), as portrayed by the schematic in Fig. 1 (see Methods). The relatively large squared Pearson correlation coefficient ($R^2 = 0.987$), small mean absolute error (MAE = 0.283) and root-mean-square error (RMSE = 0.425) of the predicted results on testing set, suggest the good generalization ability of our model (Extended Data Fig. 1). Besides, the predicted electronegativity values were also compared with those

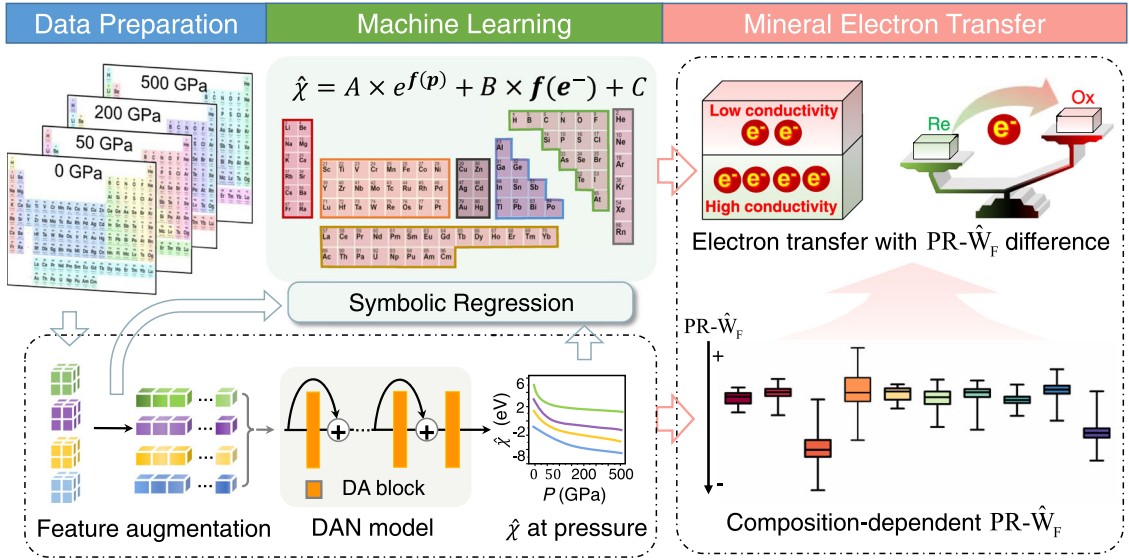

**Fig. 1 | Workflow designed to build deep learning and symbolic models for predicting electronegativity ($\hat{\chi}$) and the pressure-modulated relative work function (PR − $\hat{W}_F$).** Firstly, augmented-descriptor was generated using operation augmentation and symbolic transformer methods. Then, deep learning model Dense Attention Network (DAN) made of densely connected Dense Attention (DA) blocks fitted the data. Next, a trained DAN model was used to interpolate electronegativity under different pressures, and symbolic regression was used to fit explicit analytical expressions for the data of each partition in the periodic table. The obtained unified formula is composed of one pressure-related exponential

term (denoted as $A \times e^{f(p)}$), one electronic configuration-related linear term (denoted as $B \times f(e^-)$) and one constant term ($C$). Finally, deep learning model was utilized to calculate PR − $\hat{W}_F$ of 5828 minerals, showing they have a strong compositional dependence (ten composition-related classifications are labeled by bins in different colors). Box-and-whisker plots show the median (central line), 25th–75th percentile (bounds of the box), and maximum–minimum values (whiskers). Wok function difference can be employed in evaluating the tendency and direction of electron (e⁻) transfer, for example, between oxidants (denoted as Ox) and reductants (denoted as Re).

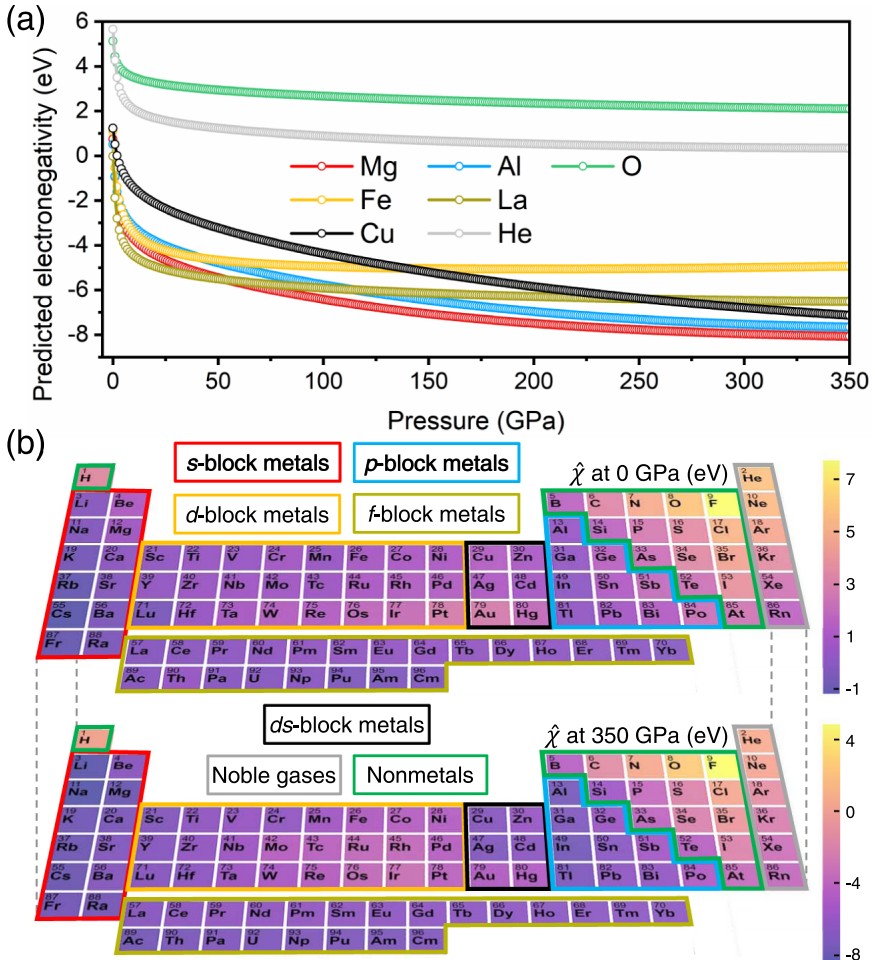

**Fig. 2 | The predicted electronegativity ($\hat{\chi}$) of elements in seven blocks varied with pressure. a** The predicted electronegativity of seven representative elements belonging to different blocks as the function of pressure (0 to 350 GPa with a 1 GPa step). **b** The predicted electronegativity ($\hat{\chi}$) at 0 and 350 GPa in the periodic table. The periodic table is colored according to the magnitude of $\hat{\chi}$. Elements in the periodic table are divided into seven blocks ($s$-block metals, $p$-block metals, $d$-block metals, $f$-block metals, $ds$-block metals, noble gases, and nonmetals) based on their electronic configuration.

DFT-calculated electronegativity of Li, C, N, Na, Mg, Ni and Au at 20, 40, 60, 80, and 100 GPa (Extended Data Fig. 2). Although the dataset used in deep learning model was not from above seven elements at five pressure points, the high prediction ability (RMSE=0.367, MAE = 0.115 and $R^2$ = 0.990) demonstrated the deep learning model had an excellent generalization performance.

The predicted results showed that the electronegativity (denoted as $\hat{\chi}$) of all elements decreased with increasing pressure (some representative elements are shown in Fig. 2a), which is ascribed to the repulsive of nuclei against compression, making it easier to remove electrons from neutral atoms (i.e., reduced electronegativity)[4,17]. The $\hat{\chi}$ values of the seven representative elements belonging to different blocks of the periodic table are still based on the homogeneous electron gas model with enthalpy as the relevant thermodynamic potential. In the first dozen GPa, they display a sharp decline trend as the pressure increases. And then as the pressure continues to rise, they show a gentle decay trend. Using nonmetal (O) as the reference element, the $s$-block (Mg), $p$-block (Al) and $ds$-block (Cu) metals keep a much faster deceasing trend of $\hat{\chi}$ at high pressure above 50 GPa, while noble gas (He), $d$-block (Fe) and $f$-block metals (La) keep similar trends. This implies the declining rate of $\hat{\chi}$ is pressure-dependent, and is also closely related with the electronic configuration of atoms. The $s$-block, $p$-block and $ds$-block metals are more sensitive to pressure. At high pressure (for example, 350 GPa), element $\hat{\chi}$ in the periodic table was found to present similar partitioned pattern as it presents at ordinary pressure (Fig. 2b). Compared to transition metals located in the middle of the periodic table with moderate $\hat{\chi}$ ($-3$ to $-7$ eV), the more oxidizing elements in the upper right (such as F and O) have larger $\hat{\chi}$ (mostly over $-1$), while the more reducing elements (such as Rb and Cs) exhibit opposite behaviors. Those $d$-block metals with more $d$ electrons like $d^5$, $d^6$, $d^7$, and $d^8$ configurations have the $\hat{\chi}$ as small as nonmetals. It can be concluded that the elements with lower electronegativity at 0 GPa also show a more significant decrease in electronegativity when compressed, indicating the electronegativity of elements with different electronic configurations varies to different degrees under pressure.

## Electronegativity as a function of pressure and electronic configuration

To deeply reveal the relationship between elemental $\hat{\chi}$ with pressure and electronic configuration, we used symbolic regression as a supervised machine learning method to obtain the empirical formula named pressure-modulated electronegativity's formula (see Fig. 1 and Methods). The formulas that were individually fitted by elements in different blocks (Fig. 2b) were obtained based on the consideration of both the simplicity and fitting accuracy (RMSE < 0.89, MAE < 0.75, $R^2$ > 0.91, Extended Data Fig. 3). All obtained formulas present the same form made of three addends (Eq. 1), including one pressure-related exponential term (denoted as $A \times e^{f(P)}$), one electronic

**Table 1 | The recommended values of constants (*A*, *B*, *C*) and functions of (*f*(*P*), *f*(e⁻)) in the pressure-modulated electronegativity's formula for each element block in the periodic table**

| Blocks | *A* | *f*(*P*) | *B* | *f*(e⁻) | *C* |
|---|---|---|---|---|---|
| *s*-block metals | 7.112 | −0.043*P* | 1.020 | $n_e$ | −8.851 |
| *p*-block metals | 7.171 | −0.019*P* | 0.980 | $n_e$ | −8.272 |
| nonmetals | 5.146 | −0.019*P* | $1.080^{n_e}$ | $n_e$ | −5.682*ln*(*n*) |
| *d*-block metals | 5.857 | −0.037*P* | 0.376 | $n_e$ | −6.671 |
| *f*-block metals | 6.693 | −0.037*P* | −0.013 | *Z* | −5.518 |
| Noble gas | 6.359 | −0.025*P* | −0.704 | *n* | 0.383 |
| *ds*-block metals | −1.628 | 0.265*ln*(*P*) | 0.451 | *n* | −0.674 |

*A*, *B*, *C*, $n_e$, *Z*, and *n* in italic are all scalar variables.

The symbol $n_e$, *Z*, and *n* represents the number of valence electrons, atomic number and the principal quantum number, respectively.

configuration-related linear term (denoted as $B \times f(e^-)$) and one constant term (*C*), respectively. The specific values for coefficients *A*&*B*, constant *C* and functions of *f*(*P*)&*f*(e⁻) are listed in Table 1.

$$\hat{\chi} = A \times e^{f(P)} + B \times f(e^-) + C \quad (1)$$

The $A \times e^{f(P)}$ term for all seven blocks are all exponential functions with pressure (*P*) as the only variable in the exponent. The negative coefficient in the linear function *f*(*P*) for the first six blocks (from −0.043 to −0.019), and in $A \times e^{f(P)}$ for *ds*-block metals (−1.628) all demonstrate the negative correlation between electronegativity and pressure. This mathematically proves that electronegativity decreases exponentially with increasing pressure, but to different degrees according to their positions in the periodic table. Particularly, the *f*(*P*) coefficient of *d*-block and *f*-block metals (−0.037), nonmetals and *p*-block metals (−0.019) is same, indicating their electronic properties have extremely high similarity. This accords with the generally accepted properties of materials, proving the rationality and accuracy of our regression formula.

Besides pressure, valence electrons (denoted as $n_e$) in *s*, *p* and *d* orbits can also change $\hat{\chi}$, where more $n_e$ leads to a larger $\hat{\chi}$, according well with the direct observation in the periodic table (Fig. 2b). Also, the $\hat{\chi}$ of nonmetals, noble gas elements and *ds*-block metals all depend on the principal quantum number (i.e., the period, denoted as *n*), while more layers of electrons for nonmetals and noble gas elements instead lead to a smaller $\hat{\chi}$. Furthermore, the *f*-block metals are only elements relating with atomic number (denoted as *Z*). Notably, a larger *Z* can result in a smaller electronegativity for *f*-block metals, such a negative correlation can also be found in their other properties such as the well-known radius lanthanide contraction. The negative constant term (*C*) also impacts on $\hat{\chi}$, whose two minimum values belong to *s*-block and *p*-block metals (−8.851 and −8.272, respectively). It indicates that under the same pressure, they always keep a smaller $\hat{\chi}$ than the other elements.

According to the changing pattern of $\hat{\chi}$ in different blocks (Fig. 2 and the pressure-modulated electronegativity's formula), it can be concluded that when the atom has fewer valence electrons in the outermost shell such as less than two *s* electrons for alkali, alkaline earth metals, copper-family and zinc-family metals, less than three *p* electrons for *p*-block metals, less than five *d* electrons for *d*-block metals, the increase in pressure leads to a more remarkable decease in $\hat{\chi}$. More valence electrons in *s*, *p* and *d* orbits all can give rise to a larger $\hat{\chi}$ (see the pressure-modulated electronegativity's formula). This could be ascribed to the metals with more electrons in *p*- and *d*-block have stronger orbital localization. High pressure makes delocalized electrons (*s* and fewer *p*, *d* valence electrons) to have a stronger ability to flow out, thus presenting a smaller $\hat{\chi}$. Hydrogen atom (H) seems to

be an exception, as its *s* electron doesn't result in a small $\hat{\chi}$ with increasing pressure, instead it behaves more like a nonmetal. Actually, according to the pressure-modulated electronegativity's formula, $\hat{\chi}$ of H is mostly intermediate between nonmetals and metals, enabling its flexible ability to lose or gain electrons. However, when faced with other metallic elements at high pressure, H is preferred as an oxidant due to its weak dependency of decreased $\hat{\chi}$ on pressure (Fig. 2b). Based on these mathematical formulas, therefore, the changing rules controlling electronegativity of individual elements at different pressures no longer require extensive and systematic calculations, which can be quantificationally and physically interpreted in a more comprehensible way. It should be pointed out that the regressed pressure-modulated electronegativity's formula and all predicted electronegativity values are based on homogeneous electron gas model and the enthalpy as the relevant thermodynamic potential, while the electronegativity in other reference systems can give different but completely comparable results.

## Pressure-induced high conductivity interpreted by mineral work functions

For solid minerals, it is no longer suitable to use element electronegativity to evaluate the overall level of electronic states. Instead, work function is highlighted since it takes orbital hybridization-induced band structure into consideration[11,12]. One of the most classical ways to estimate the work function of a compound is to calculate the geometric mean of the Pearson absolute electronegativity values of its constituent atoms[21,22]. To avoid imaginary numbers due to pressure-induced negative values in the prediction, the arithmetic mean of the predicted electronegativity values of constituent atoms was used to act as the predicted work function, named pressure-modulated relative work function and denoted as $PR - \hat{W}_F$. In fact, this calculation method gives the mean electronegativity of the whole mineral, in which the number of constituent atoms is normalized to 1. Only in this way, it is reasonable to compare the work function values of different minerals as the function of pressure, because it completely offsets those differences from pressure-induced differences in energy zero points. This estimation actually yields the relative work function, which showed approximate linearity with the experimental values (Extended Data Fig. 4). Since $PR - \hat{W}_F$ is derived from element electronegativity, it also depends on the selected reference system. In practice, we should use its relative value for comparison rather than the absolute value, because the real work function is extremely hard to measure. Considering that work function is a critical parameter related to the average oxidation state of constituent atoms[23], it is feasible and effective to employ $PR - \hat{W}_F$ as a new metric to evaluate the redox state of a single mineral phase or to predict the tendency and direction of electron transfer between different minerals under the same pressure.

Actually, there are many structural factors besides chemical composition that can alter the specific value of the work function. To this end, density functional theory (DFT) was used to investigate the impact of crystal structure and exposed surfaces on the calculated work function (Extended Data Fig. 5). It is found that work function values of different minerals with the same chemical composition but different crystal structure or exposed surfaces are still similar, with a standard deviation under 0.6 eV. Moreover, different chemical composition can cause significant work function changes, and we can predict its relative magnitude well by using $PR - \hat{W}_F$. Therefore, even though crystal structure and surface electronic state do affect the Fermi level position and thus the work function, the constituent atoms and their stoichiometry have a dominant influence when we compare different compounds and reveal general rules.

The $PR - \hat{W}_F$ values of 5828 currently approved minerals were calculated and illustrated by their categories in Fig. 3a. At normal

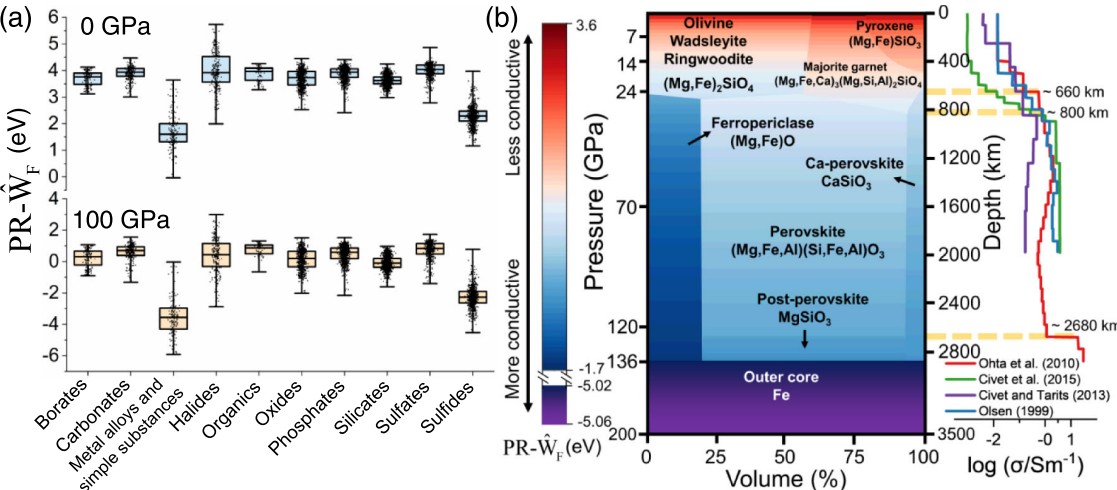

**Fig. 3 | The pressure-modulated relative work function (PR − $\hat{W}_F$) of minerals varied with mineral classifications, pressure, and depth. a** PR − $\hat{W}_F$ values of 5828 currently approved minerals at 0 and 100 GPa. Each mineral in corresponding categories is represented as black dot in the box chart. The mineral compositions and classifications are referred to the Mindat database (https://www.mindat.org/). Box-and-whisker plots show the median (central line), 25th–75th percentile (bounds of the box), and maximum−minimum values (whiskers). **b** PR − $\hat{W}_F$ of minerals in the pyrolite model with varying depth and pressure. The conductivity (denoted as σ in S/m) profiles from some references are plotted to compare with the change of work function. Shown in yellow dashed lines are the regions of high-conductivity anomaly.

conditions (0 GPa), halides have the highest PR − $\hat{W}_F$ values, with F-bearing minerals responsible for the high values. All O-dominated minerals like borates, carbonates, oxides and silicates, have similar PR − $\hat{W}_F$ because of their high content of O. Sulfides have the second smallest PR − $\hat{W}_F$ values, while pure metals and metal alloys have the smallest values. Moreover, PR − $\hat{W}_F$ values of halides, sulfides and metal alloys, as well as simple substances, are more dispersed due to the different electronegativity of their nonmetals. Although the distribution range of PR − $\hat{W}_F$ for 1540 silicates is less, the presence of some elements in silicates, especially alkaline earth metals like Ca and Mg, decrease PR − $\hat{W}_F$, which is in good agreement with the prediction of the pressure-modulated electronegativity's formula. We compared PR − $\hat{W}_F$ of all minerals at 100 GPa with those at 0 GPa, and found that the increase in pressure caused a significant decrease of ~4 eV for PR − $\hat{W}_F$ of almost all minerals (Fig. 3a). In particular, PR − $\hat{W}_F$ of metal alloys were reduced by more than 5 eV, and the maximum decrease reached ~6.3 eV for a dozen Al-bearing metal alloys. It strongly suggests that high pressure makes the mineral more reduced and, to some extent, more metallized due to the weaker binding energy of electrons. The fastest dropped PR − $\hat{W}_F$ of these minerals is closely related to the delocalization of Al $p$ electrons under high pressure. In deep regions, the only well-recognized and abundant metal alloys should be Fe and Ni, which exist in Earth's core with pressures exceeding ~136 GPa. Thus, their PR − $\hat{W}_F$ values are relatively lower than those of overlying silicates at the core-mantle boundary.

According to the well-known pyrolite model, the variations of mineral phases and phase proportions with increasing depth and pressure are shown in Fig. 3b. PR − $\hat{W}_F$ continuously decreases with pressure in the stable zone of each mineral phase, and particularly, PR − $\hat{W}_F$ drops sharply at the phase interface (Fig. 3b). Essentially, compounds with lower work function would contain more free electrons due to the weaker electron binding energy, implying that the materials from the Earth's crust to the core generally have increased conductivity. More importantly, those interfaces with large contrast in chemical compositions and work function may have a jump in conductivity. The above inferences accord well with the current magnetotelluric surveys[24–27], that the conductivity in the Earth's interior almost increases with depth, and the high-conductivity anomalies exactly locates at the boundaries like the 660 km discontinuity[28] and core-mantle boundary[26,28] (Fig. 3b). In particular, the 660 km

discontinuity between upper ringwoodite ((Mg,Fe)$_2$SiO$_4$) and lower ferropericlase have -0.9 eV of work function difference (denoted asPR − $\Delta\hat{W}_F$) and the nearly double enhancement of conductivity. Similar interface was also discovered between majorite and underlying ferropericlase at -800 km with ~1.2 eV of PR − $\Delta\hat{W}_F$ and a three-fold increase of conductivity.

Notably, the conductivity at the core-mantle boundary is improved by more than an order of magnitude, not only due to the existence of Mg in the post-perovskite with lower PR − $\hat{W}_F$ than Fe-bearing perovskite, but also the formation of ohmic contact-like interface with extremely large PR − $\Delta\hat{W}_F$ (-4.4 eV) that can drive the directional flow of electrons from the Fe outer core (PR − $\hat{W}_F$ as low as −5.0 eV) to the upper silicates. Therefore, PR − $\Delta\hat{W}_F$ at the interface is proportional to the changes of deep conductivity to a certain extent. The reason for such a dramatical decrease in PR − $\hat{W}_F$ of the underlying minerals is the loss of nonmetal O and Si and the enrichment of low-electronegativity metal like $s$-block Mg and $d$-block Fe, as predicted and interpreted by the pressure-modulated electronegativity's formula. Based on this, it can be further assumed that the existence of a large amount of complicated phase interfaces in the heterogeneous deep Earth may be one of the reasons for the conductivity inhomogeneity[28].

## Work function-informed redox reactivity between Fe(II)-bearing minerals and water under high pressure

For systems with different components, pressure has different degrees of influence on the energy[3], in which the work function is one significant quantitative index. At the mineral phase boundary, electrons will spontaneously flow from the phase with lower work function to the other[29,30], thus driving a possible redox reaction. Inside the Earth, the reactivity of minerals with water is vital to our understanding of deep element cycle[6,31,32]. Actually, natural oxides and silicates account for more than 90% of the total weight of Earth's crust. Most of them known as hydrous minerals (such as mica and hornblende) and nominally anhydrous minerals (such as pyroxene and quartz), contain up to a few hundred parts per million $H_2O$ in form of OH group, especially in the mantle transition zone[16,33–35]. Since silicates are the essential rock-forming minerals, the reactivity between Fe(II)-bearing silicates and surrounding water through electron transfer-dominated redox reactions during rapid deep subduction deserves to be pondered

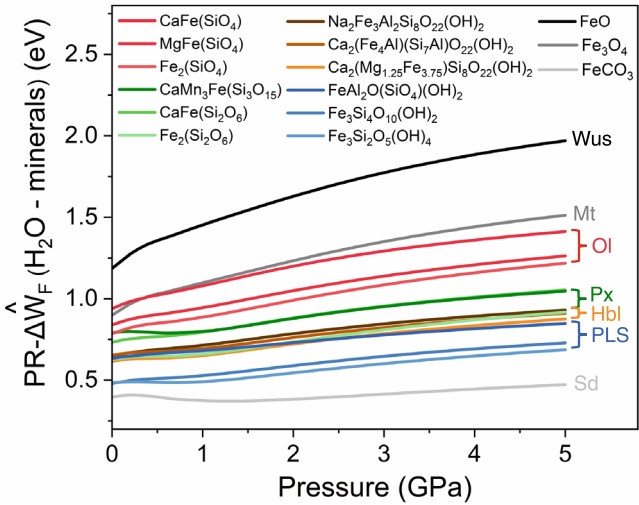

**Fig. 4 | The pressure-modulated relative work function difference (PR − $\Delta\hat{W}_F$) between Fe(II)-bearing minerals and water as the function of pressure.** The composition of silicate minerals refers to International Mineralogical Association (IMA) database (https://rruff.info/ima/). The symbol "Wus", "Mag", "Ol", "Px", "Hbl", "PLS", "Sd" represents wüstite, magnetite, olivine, pyroxene, hornblende, phyllosilicate, siderite, respectively.

deeply. We calculated the PR − $\Delta\hat{W}_F$ between $H_2O$ and 12 representative silicate minerals as well as wüstite, magnetite, siderite up to 5 GPa (Fig. 4). The relative magnitude of PR − $\Delta\hat{W}_F$ suggests that the chemical reactivities of Fe(II)-bearing silicate minerals with $H_2O$ increases with pressure, and follows the order of olivine > pyroxene > hornblende > phyllosilicate. The predicted order of activity: magnetite > fayalite > hedenbergite, accords well with the experimental results of $H_2$ production[31]. Under less high pressure, $H_2$ would be the only possible product rather than hydrides when $H_2O$ accepts electrons. Predictably, olivine and its high-pressure polymorphism wadsleyite and ringwoodite may perform high reactivity with $H_2O$ and $H_2$-producing activity up to the 660 km discontinuity (~24 GPa). Such a prediction has been verified by a recent study that directly observed inclusions from the lower mantle containing ringwoodite, OH groups and $H_2$ molecules[35]. Based on DFT calculations, it is found that oxides and silicates minerals that are superficially bonded with protons (i.e., forming adsorbed OH groups) would have considerably decreasing work function by more than 2 eV (Extended Data Fig. 6). Such a change in work function actually comes from the change in the surface molecular environment and electronic state, suggesting the robust electronic interaction between metal-bearing minerals and surface OH groups.

In most cases, OH groups are directly bonded to metals such as Fe, Mg, Ca even at dozens of gigapascals[36]. According to the pressure-modulated electronegativity's formula, with the increase of pressure, the electron transfer tendency from metal to oxygen is strengthened due to the increasing $\Delta\hat{\chi}$ between metal and oxygen. So, it is the pressure that induces the non-uniform changes in the electronic states of different atoms[3,4], which could expand the degree of overlapped density of state and thus enhance the metal-to-ligand (OH) electron transfer. Consistent with Marcus' electron transfer theory[2], the reaction rate of electron transfer between the electron donor and acceptor is proportional to the square of electronic coupling, which strongly depends on the two geometric factors affecting the overlap of electronic wavefunction, i.e., their distance and orbital orientation. Although the orientational dependence on pressure is unknown, it is easy to conclude that pressure could promote reaction rates at least from the perspective of enhancing the overlap of electronic wavefunction through shortening the distance between donor and

acceptor. As a result, the reduced distance between metal and bonded OH group under pressure facilitates the electron transfer between them, which might bring about dehydrogenation, following $H_2$ production and even decomposition of hydrous silicate minerals[37,38].

Furthermore, according to the pressure-modulated electronegativity's formula, we propose that some common transition metals like Ni and Co substituting Fe would increase the whole PR − $\Delta\hat{W}_F$ of Fe(II)-bearing minerals because of the more *d* valence electrons in Ni and Co than Fe. Instead, the improvement in redox reactivity between Fe(II)-bearing silicates and water can be found in the substitution of Al for Si, ascribed to the highly delocalized valence electrons of Al at high pressure as discussed above. It is thus concluded that Fe(II)-bearing minerals, although some of them were rarely mentioned in research works, may be robust $H_2$-producing agents and significant sources of mantle $H_2$ under sufficiently high pressure (for example, in subduction processes). These predicted reactions may provide a new pathway to produce mass $H_2$ throughout the deep Earth, thus playing considerable roles in controlling oxygen fugacity. Moreover, the pressure-induced redox reactions may also occur between metal-bearing minerals (for example, Fe(II)-bearing silicate minerals) and inorganic carbon (for example, carbonates or carbon dioxide) with the increasing depth in Earth's interior, and give rise to methane and other organics, due to the increasing work function difference between them. The above inference has been verified through the direct observation from the deep inclusions, in which there were reduced abiotic products such as methane, diamond in the process of low-temperature cold subduction[39,40]. It should be noted that temperature can not only greatly alter the concentration and state of surface adsorbed species, but also effectively promote the reaction rate according to Arrhenius equation. Further work related to temperature effect on electronegativity and work function deserves more theoretical and experimental efforts. But considering the relativity of electronegativity and work function, the comparison between different minerals under the same conditions (i.e., same pressure or temperature) is of significant practical value.

## Further implications

Until now, we know very little about the behavior and mechanism of electron transfer in the deep regions of Earth's interior. Full appreciation of the Earth's deep redox processes and the ways in which they influence the element cycle of Earth's interior, require further unveiling of the electron transfer rules among minerals, fluids, melts and volatiles. Our study suggests the weakening binding energy of electrons in individual atoms and the enlarged interfacial work function difference between metals and nonmetals under pressure, greatly activate the electron transfer reactions in the deep Earth. Note that the element electronegativity is an essential parameter to correlate with bond energy, covalent radius and polarization and so on, the dependence of electronegativity on pressure and electronic configuration shown in our predicted formula therefore could be extended to understanding other thermodynamic properties (such as formation energy and ionization energy). Improved applications of pressure-induced electron transfer can also be further developed in other fields, including but not limited to batteries, electrocatalysis, sensors and electromicrobiology.

## Methods
### Electronegativity data preparation and deep learning method
**Element electronegativity dataset.** The element electronegativity dataset includes 96 element electronegativity (from H to Cm), calculated by the density functional theory (DFT) method under 4 pressure values (0, 50, 200, and 500 GPa)[17]. The dataset containing only $96 \times 4 = 384$ data points is slightly small, which might lead to overfitting in deep learning models.

To diminish the impact of small dataset and build reliable deep learning models, it is important to design relevant features that collectively capture the electronegativity trends across different elements and pressure values. The features should be discriminative enough to uniquely represent an element. Thus, in line with the definitions and properties of an element, we design three sets of feature descriptors: (1) the electronic structure set, to represent valence electron of elements; (2) the physical property set, to represent element physical properties; and (3) the periodic table location set, to describe the position of element in the periodic table. Different subtypes of descriptors in each set, along with their counts, are provided in Extended data Table 1. Considering the particularity of lanthanide and actinide metals (such as lanthanide shrinkage), we additionally introduce pseudo row number and pseudo group number to encode the position of element in the periodic table. Pseudo row number is the standard row number except for the lanthanoids and actinoids, which are 8 and 9, respectively. Pseudo group number refers to the standard group number except for the lanthanoids and actinoids either. It encodes La-Lu and Ac-Lr from 3 to 17. To sum up, the elemental electronegativity dataset consists of 384 data points, where each composes of 12-dimensional input features (11-dimensional element descriptor shown in Extended data Tables 1 and 1-dimensional pressure) and 1-dimensional output electronegativity.

**Feature augmentation.** The distribution of input features has a great impact on the performance of neural network. In order to reasonably fit data with deep learning network, features were augmented through operation augmentation and symbolic transformer. The feature augmentation workflow was presented in Extended Data Fig. 7. Operation augmentation, using groups of functions to generate augmented features, was performed on the initial 12-dimensional features. In this paper, square, logarithm, square root and reciprocal operations, represented by $x^2$, $\log(|x + 1.0|)$, $\sqrt{|x| + 1.0}$, $\frac{1}{x+1.0}$, respectively, were used to generate another four augmented features for each original feature and thus increase the dimensionality of features from 12 to 60. Then symbolic transformer[41] was performed on the 60-dimensional augmented features. The process of the symbolic transformer is as follows. Firstly, a set of compact closed-form formulas using symbolic regression[42] is established, which uses genetic algorithm to combine algebraic expressions stochastically. Secondly, suitable formulas are screened out based on Pearson correlation coefficient between the predicted value of formula and the ground truth. Thirdly, in order to avoid generating collinear features which make the model training process unstable, those formulas with the least correlation to one another among the suitable formulas are added as new features. In this work, 200 suitable formulas which are linear combination of 60 augmented features were screened out as candidates. Finally, 10 formulas with the least correlation to one another among 200 suitable formulas were picked up as new features. By this way, the dimensionality of features was increased from 60 to 70 after symbolic transformer method.

**Dense attention network (DAN).** To tackle the problem of small dataset which might lead to overfitting, we proposed Dense Attention Network (DAN), a specialized architecture to learn from data with small size. As shown in Extended Data Fig. 7, DAN leverages "dense connection" and "self-attention" mechanisms to overcome the difficulties of training on small data. First, DAN composes of densely connected Dense Attention (DA) blocks, which encourage feature reuse and propagation. Furthermore, it is observed that dense connection has a regularization effect[43], which reduces overfitting on tasks with small training set. Second, each DA block consists of a trunk network and a branch network. The trunk network is a 3-layer feed-forward neural network, in which the first two layers use multi-head self-attention module (MSA) for embedding. MSA is good at capturing the intrinsic relevance of features and choosing salient features to reason, which improves model performance[44]. The last layer of the trunk network uses linear projection operator to fuse the embedding vectors and output 1-dimensional predictions. The branch network of DA block is directly connected to the output predictions via a linear skip-connection, which is an effective technique to improve the performance and convergence of deep neural network. Linear skip-connection can relieve the difficulties in optimization due to nonlinearity by propagating a linear component through the neural network layers[45].

Hyperparameters of DAN, including dimensions of embedding tensor, learning rate and batch size, were determined using Bayesian optimization[46], which is a state-of-the-art hyperparameters optimization method. The details of hyperparameters settings are presented in Extended data Table 2.

The element electronegativity dataset was split into training set and testing set with the ratio of 9:1. We train DAN using Smooth $L1$ loss shown in Eq. 2, in which $y_{pred}$ is DAN predicted element electronegativity and $y_{gt}$ is the corresponding ground truth. The definition of smooth $L1$ function is shown in Eq. 3.

$$L = \text{Smooth}_{L1}(y_{pred} - y_{gt}) \tag{2}$$

$$\text{Smooth}_{L1}(x) = \begin{cases} 0.5x^2 & \text{if } |x| < 1 \\ |x| - 0.5 & \text{otherwise} \end{cases} \tag{3}$$

To further alleviate the problem of easy overfitting due to small training set, we also used $k$-fold bagging ensemble method to reduce the variance of models[47]. $K$-fold bagging method separates the training set into $k$ folds to train $k$ base models ($k - 1$ fold for training and 1 fold for validation) and ensembles $k$ base models by averaging the output of the base models. In this work, we set $k = 5$ and trained 750 epochs for each DAN base model. During training the base model, we further adopted data augmentation (Extended data Table 2) through adding Gaussian noise to input features and labels to avoid overfitting of the base model. Also, the strategy of early stop was used, so that the overfitting would be alleviated.

**Model performance and ablation studies.** In this work, root-mean-square error (RMSE), mean absolute error (MAE) and squared Pearson correlation coefficient $R^2$ were used to assess the performance of element electronegativity regression task. Baseline methods include TabNet[48], NeuralNetFastAI[47], CatBoost[49], LightGBM[50], XGBoost[51] and Random Forest[52]. The comparison results of the element electronegativity regression models are summarized in Extended data Table 3. DAN model achieves *RMSE* 0.425, *MAE* 0.283, and $R^2$ 0.987 in the testing set, which are the best among those methods. This indicates that our proposed model DAN has good generation performance in the task of element electronegativity prediction, thus obtaining more accurate work function calculation results.

In order to further verify the effectiveness of our proposed training strategies, ablation experiments were performed and the results are listed in Extended data Table 4. We take 3-layer Multilayer Perceptron (MLP) with linear skip-connection as a baseline and gradually add the above strategies to see their impact on the model performance. As shown in Extended data Table 4, feature augmentation methods (operation augmentation and symbolic transformer), model mechanisms (self-attention and dense connection) and ensemble learning method (5-fold bagging) all have a positive effect on model performance improvement.

**Element electronegativity symbolic model synthesis**
After training the element electronegativity prediction DAN model, a black box function $\phi$ that can predict the electronegativity of elements

under different pressure values was obtained. In order to reveal the influence of element descriptors and pressure on electronegativity, we used the symbolic regression package *PySR*[53] to fit closed-form analytical expressions to $\phi$. The analytical expression fitting process is explained in details as follows. Firstly, black box function $\phi$ is used to interpolate the electronegativity under different pressure values. Secondly, the periodic table are partitioned into several blocks (*s*-block metals, *d*-block metals, *f*-block metals, noble gas, nonmetals, *p*-block metals and *ds*-block metals) according to valence electrons, since element electronegativity is strongly dependent on the valence electrons (Fig. 2b). Finally, symbolic regression, a supervised machine learning method[42], was used to fit analytical expressions to the data of each partition and substituted the expressions into the model to create an alternative symbolic model.

Analogous to Occam's razor, the "fitness" of each expression is defined in terms of simplicity and accuracy. The process of selecting the best analytical expression[53] is as follows. Firstly, multiple candidate analytical expressions are provided at different complexity levels, where complexity is scored as the number of operators, constants and input variable. The simplicity and accuracy of each formula are measured by complexity and mean square error (MSE), respectively. Formulas with higher complexity and MSE will be screened out. Secondly, formulas are sorted in ascending order by complexity. The "fitness" of each formula is measured by fractional drop $f_d$ in Eq. 4, in which $c_i - c_{i-1}$ represents the difference of adjacent complexity and $\log(MSE_i - MSE_{i-1})$ is the difference of corresponding logarithm of MSE. Finally, the analytical expression that maximizes the fractional drop is considered as the best formula. The performances of analytic expression corresponding to each block are shown in Extended Data Fig. 3.

$$f_d^{\,i} = -\frac{\log(MSE_i) - \log(MSE_{i-1})}{c_i - c_{i-1}} \qquad (4)$$

Symbolic regression uses genetic algorithms to assemble analytical functions, which is essentially a brute force procedure, so the symbolic model search space scales exponentially with the number of input variables and operators. To reduce the search space of symbolic models, the input variables considered in the symbolic model fitting process are only 11-dimensional original element descriptor (Extended data Tables 1) and 1-dimensional pressure rather than 70-dimensional augmented features. The operators considered in the symbolic model fitting process are addition, subtraction, multiplication, division, logarithm and exponent.

### Computational details of deep learning
All the deep learning experiments were performed on one NVIDIA RTX 2080Ti graphics processing unit with 11GB of memory. We write our DAN with PyTorch and train it using Adam optimizer[54]. The package *PySR*[53] is used to model the output data of our DAN through symbolic models.

### Work function calculation based on density function theory
The work function was calculated using the VASP 6.2.1 code[55,56]. The plane-wave basis set was applied in the framework of the projector augmented wave (PAW) method[57]. The exchange-correlation energy was determined using the generalized-gradient approximation (GGA) defined by Perdew and his coworkers[58]. Valence electrons included O $2s^2 2p^4$, Si $3s^2 3p^2$, S $3s^2 3p^4$, Al $3s^2 3p^1$, Zn $3d^{10} 4s^2$ and Ti $3p^6 3d^2 4s^2$ was chosen. The DFT + U method with $U-J = 4.2$ eV was used to describe the strong correlation of the localized Ti 3d states[59]. According to previous researches, the most stable crystal surfaces for $TiO_2$[60], $ZnS$[61] and $Al_2SiO_5$[62] were selected. According to the size of crystal lattice, 4–8 layers thickness were chosen. The vacuum layer thickness was chosen to be 20 Å to avoid the effects of periodic structures. The atoms within the range of the bottom layer of 5 Å were fixed to simulate the bulk

phase structure. The K-point was generated by VASPKIT 1.3.0[63] code and K-mesh Resolved Value was set to 0.04 by Gamma Scheme. All structures were fully relaxed until the residual force converged to less than 0.02 eV/Å. The energy cutoff was set was to 400 eV. The total electronic energy converged to less than $10^{-6}$ eV. Furthermore, the work function was calculated as the difference between the average electrostatic potential in the vacuum and the Fermi energy of the slab, which got from the static calculations and the ecteronic energy converged to less than $10^{-8}$ eV.

## Data availability
Source data are provided with this paper in the Source Data file. The collected element electronegativity dataset and the experiment data are available at https://github.com/GCaptainNemo/Electronegativity-Under-Pressure and https://doi.org/10.5281/zenodo.7709844[64].

## Code availability
The source code of Dense Attention Network and the experiment scripts are publicly available at https://github.com/GCaptainNemo/Electronegativity-Under-Pressure and https://doi.org/10.5281/zenodo.7709844[64].

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

## Acknowledgements

Sincere thanks go to Prof. Xiao Dong and Artem R. Oganov for their pioneering work (doi: 10.1073/pnas.2117416119) that provided 96 ele-ment electronegativity under 4 pressure values (0, 50, 200, and 500 GPa)[17]. This work was supported by the National Natural Science Foundation of China (grant nos. 91951114 and 41872042 to Y.L., 92251301 to A.L., 62271016 to X.B., and 41522201 to Y.L.) and the DDE-IUGS Big Science Program.

## Author contributions

Y.L., A.L., Y.Z.L,. H.W., X.B. conceived the study. H.W., Y.Z.L., Y.L., X.B., H.Y., Y.Z., R.Y. provided methodology and performed modeling. Y.Z.L., Y.L., H.W., X.B., A.L., H.J., H.Y., B.H., C.W., H.D. processed the data and interpreted the results. Y.Z.L., Y.L., H.W., Y.Z. wrote the paper. Y.L., X.B., Y.Z., A.L. contributed to paper polishing.

## Competing interests

The authors declare no competing interests.
