## [Peer Review File · Nature Communications]

REVIEWER COMMENTS

Reviewer #1 (Remarks to the Author):

The authors used the published pressure induced electronegativities to predict the work functions of many minerals and applied the database of work functions to analysis the hydrogen transfer inside the Earth. Their paper is interesting, and their ideas are innovative. However, their description and discussion need to be more precise and some of their conclusion (especially in H₂ production) could perhaps be wrong. The followings are the comments:

Electronegativities:

1. One of my main concerns is the energy zero points they used. By the definition of Mulliken electronegativities, we must supply the energy of one single electron for the electric neutrality. It is not problem at zero pressure. But at high pressure, the definition of the energy and enthalpy of single electron could be artificial. In PNAS 119, e2117416119, the authors set energy zero as free electrons at specific pressure. But this selection is not unique and natural, some other energy zero could be used. In this way, discussion of electronegativity differences at same pressure is OK but discussion of the electronegativity tendencies at different pressures artificially relies on the choice of their energy zero and could be problematic when the author extended this pressure induced tendencies further (comment #2).

Work functions:

2. Following by comment #1, since their electronegivities contain the contribution of single electron at different pressure, their work functions also contain this contribution. When the authors compared

work functions at different pressures, such as fig. 3b, their conclusion cannot exclude the contribution of the artificial enthalpy of single electron at different pressures, too. In fact, the chemical hardness is more suitable for the metallization they want to discuss.

3. Their method of “the arithmetic mean of the predicted electronegativity values of constituent atoms” is a VERY VERY COARSE method for the prediction of work function. They ignored the crystal structures, the chemical valences, charge transfer and electronic state. As they mentioned, the pyrolite has several phase transitions in the mental situation, such as pv to ppv. How do the structures affect the work function? Maybe DFT calculations of work functions with different structures are needed to verify their validity of methods.

4. Charge transfer is also an important factor for the work function. The Mulliken electronegativity rely on the charge state of the atoms and ions. The electronegativities are different for Na^+ and Na , Ca^{2+} and Ca , Mg^{2+} and Mg by the octet rule. The neglects of these facts lead to some misunderstanding conclusion, such as “ More importantly, those silicate minerals that have fewer connective silicon-oxygen tetrahedra and larger structure space to accommodate alkali metals alkaline earth metals with lower electronegativity like Ca , Mg , Na .”

Hydrogen transfer

5. Their logic connection of work function and H_2 -producing activity is not solid and need more proofs. Perhaps when H_2O meet the minerals, the charge transfer occurs by the work function difference. But the authors should prove the work function difference is big enough and the charge transfer is sufficient for the reduction of H_2O to H_2 . If the charge transfer is quite small and fail to produce H_2 , their causal relationship between work function difference and reaction to water is not solid as they claimed.

6. Following by comment #4, based on HSAB (hard soft acid base) theory, I consider the theory based on chemical hardness is more reliable, because most mineral reactions with H_2O have no charge transfer.

7. The authors claimed, “Such predictions are in good agreement with experimental observations that the metallic Fe , FeO , Fe_2O_3 and FeO_2 begin to react with H_2O at 5, 78, 110 GPa and over 130 GPa, respectively”. But by the reaction equations supplied in Fig 4a, some of the equations ($\text{Fe} + \text{H}_2\text{O} \rightarrow \text{FeO} + \text{FeH}_x$) has charge transfer and some have none ($\text{Fe}_2\text{O}_3 + \text{H}_2\text{O} \rightarrow \text{FeO}_2\text{H}_x$, in fact only $x=1$ can be balanced). The reaction mechanisms are different. Their prediction of the transfer of electrons from FeO_x to H_2O cannot explain these all.

8. Work functions in Fig4b cannot give the results about H_2 production, also cannot be a scale of acidity. OH is an unstable free radical, not a stable compound. The differences between OH and minerals lack of exact physical meanings.

Reviewer #2 (Remarks to the Author):

Comments:

The authors developed a Dense Attention Network (DAN) framework to predict the electronegativity of 96 elements under high pressure conditions. With the predicted electronegativity, the work function of 5828 minerals were further evaluated and were correlate with the electron transfer mechanism of minerals in deep Earth's interior. The introduction of the DAN algorithms was well-written, and the authors demonstrated systematic and robust methods to prevent overfitting of the DAN model. However, the description of the establishment and the validation of dataset is ambiguous. If a data set could not be well verified, the data-driven results would be unreliable. With this in mind, I believe that the following points should be addressed prior to publication to enhance the impact of the manuscripts.

Detailed comments:

1. A summary about the calculation of electronegativity via the Density Functional Theory (DFT) methods should be provided, instead of just referring to a citation (Ref. 17). Also, since there are several types of electronegativity, the authors should specify that the electronegativity in this work is the Mulliken electronegativity, and briefly comment on the difference between the Mulliken electronegativity and the well-known Pauling electronegativity.
2. The reported values of electronegativity are not consistent with the common sense about electronegativity. For instance, the electronegativity of fluorine in Extended data Fig. 2 is around seven, but the well-known value should be around four. This deviation may be attributed to the different definitions of electronegativity or the error from DFT calculations. If the author can clarify the deviation, it will avoid the misunderstanding by readers.
3. In Fig. 2a, the machine learning (ML) predicted electronegativity considerably decreases in the pressure regime from 0 GPa to 20 GPa. However, the ML data set only includes the electronegativity at 0 GPa and 50 GPa. The authors should verify that the sharp reduction of the electronegativity is not the artefact due to the lack of training data in this pressure regime.
4. The evaluation of work function based on the arithmetic mean of the electronegativity values of constituent atoms may be a reasonable strategy for pure mental elements, but it could be an oversimplified approach for the minerals with hydrogen and oxygen atoms, which are sensitive to their morphology or their molecular environments. If the authors can provide a comparison between their method and the other robust approaches for the calculation of work of some common minerals, the results and discussion will be more convincing.

5. At page 9, the authors said that "Consistent with Marcus' electron transfer theory, the electron transfer tendency in the metal-ligand bonding system strongly depends on the distance between the electron donor and acceptor." According to Marcus' theory, electron transfer rate is proportional to the square of electronic coupling, which is dependent on not only distance between the electron donor and acceptor but also their orientation. Under high pressure, as all the atoms are in close contact, the atomic or molecular alignments play a decisive role in electronic coupling that determines the electron transport rate, especially for minerals with non-metal elements. Therefore, the current description at page 9 could be misleading, and it needs to be modified.

6. The values of electronegativity and work function are also affected by the morphology or the molecular environment of minerals. In the deep regions of Earth's interior, high temperature can lead to dramatic change of the molecular environment. In this work, it seems that both electronegativity and work function are obtained from the prediction (or evaluation) of individual molecular configuration. The authors should briefly discuss about the impact of temperature and molecular environment.

Reviewer #1 (Remarks to the Author):

The authors used the published pressure induced electronegativities to predict the work functions of many minerals and applied the database of work functions to analysis the hydrogen transfer inside the Earth. Their paper is interesting, and their ideas are innovative. However, their description and discussion need to be more precise and some of their conclusion (especially in H₂ production) could perhaps be wrong. The followings are the comments:

Reply: We really appreciate your positive feedback and all professional comments. According to your suggestion, we have added more introduction to Dong's electronegativity (Dong et al., 2022) and put strict restrictions on the applicable conditions for the comparison of electronegativity between elements. Besides, we emphasized in the revised manuscript that the contribution of this work to the electronegativity of element at arbitrary pressure up to 500 GPa was obtained by both the pioneering referenced dataset and our robust machine learning method. Since the work function of minerals is just one case of element electronegativity application, we have revised this section to underline that it is the changes in the fundamental physicochemical property of elements under pressure affecting the electric conductivity and reactivity of minerals. Moreover, we added DFT calculations to show the factors influencing the work function of natural minerals are primarily the changes of chemical composition, rather than structural adjustment (crystal structure type, exposed crystal planes) under the same chemical formula, thus showing the current work function calculation method is suitable for evaluating the major trends of electron transfer in complex Earth systems.

We sincerely hope this version has improved under your insightful guidance. All revisions were highlighted in yellow in the revised manuscript. A point-by-point response to your comments is given below for your further review.

Electronegativities:

1. One of my main concerns is the energy zero points they used. By the definition of Mulliken electronegativities, we must supply the energy of one single electron for the electric neutrality. It is not problem at zero pressure. But at high pressure, the definition of the energy and enthalpy of single electron could be artificial. In PNAS 119, e2117416119, the authors set energy zero as free electrons at specific pressure. But this selection is not unique and natural, some other energy zero could be used. In this way, discussion of electronegativity differences at same pressure is OK but discussion of the electronegativity tendencies at different pressures artificially relies on the choice of their energy zero and could be problematic when the author extended this pressure induced tendencies further (comment #2).

Reply: Thank you for this very valuable and helpful comment. You kindly reminded us of the difference between Dong's electronegativity and Mulliken electronegativity (Dong et al., 2022). We totally agree with you that the direct comparisons of electronegativity values at different pressures should be avoided. As you pointed out, these data should be used carefully, so there must be some reasonable preconditions and constraints. In this regard, we made corresponding revisions to further clarify the definition of our referenced Dong's electronegativity and its difference from Mulliken electronegativity, and removed the contents that might involve direct comparison of the absolute electronegativity values of the same element under different pressures. All revisions are highlighted in yellow in the revised manuscript and copied as follows for your convenience:

“The newly-calculated electronegativity from Dong and his co-workers appropriately modified the definition of Mulliken electronegativity for better use at high pressures, and it used the enthalpy as the relevant thermodynamic potential within the framework of homogeneous electron gas model. At zero pressure, Dong's electronegativity has the same energy unit as Mulliken electronegativity (eV) and also has a good linear correlation with Pauling electronegativity (correlation coefficient R

= 0.91) (Lines 96-102 in the revised manuscript)".

"Although Dong's electronegativity dataset has different reference systems and specific data from the other two well-known electronegativity datasets, it successfully extends the applicability of evaluating electron transfer trends between elements to high pressure cases. The predicted stability of Mg_xFe_y compounds at pressures above 100 GPa accords well with the experimental observations¹⁷, indicating Dong's electronegativity after modifying Mulliken electronegativity is specifically suitable for high pressure cases (Lines 104-110 in the revised manuscript)".

Actually, as discussed above, the absolute electronegativity value is of little significance. For example, the electronegativity of F and Mg is calculated as 7.3 and 0.8 eV at 0 GPa, and 5.2 and -7.3 eV at 200 GPa, respectively (Dong et al., 2022); while the values are bound to change when using another energy zero point as reference. Therefore, the comparison of the absolute values is meaningless. However, there is no problem in comparing the change of the electronegativity difference between the two elements along with pressures, because such an electronegativity difference completely offsets the artificial factor of the choice of energy zero point at specific pressures. In the above example, regardless of which energy zero point is chosen, the electronegativity difference between F and Mg is 6.5 eV at 0 GPa and 12.5 eV at 200 GPa. In this case, we can draw a safe conclusion that the electron transfer tendency between Mg and F is stronger at 200 GPa than that at 0 GPa. According to this, we have emphasized the comparison of electronegativity difference between the two elements at different pressures rather than the change in the electronegativity of an element at different pressures: "Using nonmetal (O) as the reference element, the *s*-block (Mg), *p*-block (Al) and *ds*-block (Cu) metals keep a much faster decreasing trend of χ at high pressure above 50 GPa, while noble gas (He), *d*-block (Fe) and *f*-block metals (La) keep similar trends (Lines 151-154 in the revised manuscript)".

In other places involving the comparison of electronegativity variation trends under different pressures, we added strict restrictions to avoid misunderstanding or misuse. The specific revisions are as follows:

"The χ values of the seven representative elements belonging to different blocks of the periodic table are still based on the homogeneous electron gas model with enthalpy as the relevant thermodynamic potential. In the first dozen GPa, they display a sharp decline trend as the pressure increases. And then as the pressure continues to rise, they show a gentle decay trend (Lines 146-151 in the revised manuscript)".

"It should be pointed out that the regressed pressure-modulated electronegativity's formula and all predicted electronegativity values are based on the homogeneous electron gas model and the relevant thermodynamic potential of enthalpy, while the electronegativity in other reference systems give different but completely comparable results (Lines 220-224 in the revised manuscript)".

References:

Dong, X., Oganov, A. R., Cui, H., Zhou, X. F., & Wang, H. T. (2022). Electronegativity and chemical hardness of elements under pressure. *Proceedings of the National Academy of Sciences*, 119(10), e2117416119.

Work functions:

2. Following by comment #1, since their electronegivities contain the contribution of single electron at different pressure, their work functions also contain this contribution. When the authors compared work functions at different pressures, such as fig. 3b, their conclusion cannot exclude the contribution of the artificial enthalpy of single electron at different pressures, too. In fact, the chemical hardness is more suitable for the metallization they want to discuss.

Reply: Yes. You're absolutely right. Since the work function is the arithmetic mean of the

electronegativity values of constituent atoms, it will also be affected by the definition of energy zero point. As replied above, it is feasible to compare the electronegativity difference between the two elements at different pressures. Notably, to compare the difference in the work function between two compounds at different pressures, the premise is that the number of elements on both two sides must be the same, because only in this way can the differences of different energy zero points caused by pressure be offset. So, we made revisions on the discussion of the work function by comparing the work function difference between the two compounds at different pressures. For your convenience, the detailed revisions are as follows:

“In fact, this calculation method gives the mean electronegativity of the whole mineral, in which the number of constituent atoms is normalized to 1. Only in this way, it is reasonable to compare the work function values of different minerals as the function of pressure, because it completely offsets those differences from pressure-induced differences in energy zero points (Lines 240-244 in the revised manuscript)”.

On the question of chemical hardness, since the solid-state analogs of Mulliken electronegativity and chemical hardness are equal to the work function and band gap, respectively, it is indeed more reasonable to link metallization with chemical hardness. What we actually found was the pressure-induced decreases in the work function of minerals, suggesting that the electrons in the minerals have weaker binding energy and stronger escape capability under high pressure. Such a change is not only due to the increased metallization, but may also be due to the intensifying electron-donating reactivity. Therefore, to avoid misunderstanding or misuse, we have deleted those discussion about metallization because it is less relevant to our main theme.

3. Their method of “the arithmetic mean of the predicted electronegativity values of constituent atoms” is a VERY VERY COARSE method for the prediction of work function. They ignored the crystal structures, the chemical valences, charge transfer and electronic state. As they mentioned, the pyrolite has several phase transitions in the mental situation, such as pv to ppv. How do the structures affect the work function? Maybe DFT calculations of work functions with different structures are needed to verify their validity of methods.

Reply: Many thanks for this comment and constructive suggestion. As described in the manuscript, one of the most classical ways to estimate the work function of a compound is to calculate the geometric mean of the absolute electronegativity of its constituent atoms according to the chemical formula (Nethercot, 1974; Chen et al., 1977; Xu and Schoonen, 2000; Radha et al., 2019; Naseri et al., 2021). The work function values estimated by this method accord well with the experimental values of dozens of compounds (Chen et al., 1977). However, the electronegativity values of elements are generally negative under high pressure, so the work function calculation method based on geometric mean value is no longer applicable. Moreover, as replied in comment #2, to compare the difference in work function between two compounds at different pressures, the number of elements on both two sides must be the same, because only in this way can it offset the differences in different energy zero points caused by the pressure. To avoid the above problems, we used the arithmetic mean of electronegativity of constituent atoms to represent relative work function, which has the comparative significance of relative values between different compounds, while its absolute value doesn't make sense. We calculated the relative work functions of some compounds for which the experimental values were available in the literature, and found that they were approximately linearly related with the experimental values (Fig. R1-1). So, note that the calculated work function in this work can reflect relative magnitudes, it is applicative in those cases where they need to evaluate the electron transfer tendency and direction under the same pressure. To distinguish our calculated relative work function from the absolute or experimental one, we named it as pressure-modulated relative work function (denoted as $PR-\overline{W}_F$) in the revised

manuscript.

We made corresponding modifications in the text and copied them as follows: “This estimation actually yields the relative work function, which showed approximate linearity with the experimental values (Extended Data Fig. 4). Since $PR-\widehat{W}_F$ is derived from element electronegativity, it also depends on the selected reference system. In practice, we should use its relative value for comparison rather than the absolute value, because the real work function is extremely hard to measure. Considering that work function is a critical parameter related to the average oxidation state of constituent atoms²³, it is feasible and effective to employ $PR-\widehat{W}_F$ as a new metric to evaluate the redox state of a single mineral phase or to predict the tendency and direction of electron transfer between different minerals under the same pressure (Lines 244-253 in the revised manuscript)”.

Fig. R1-1 Comparison of calculated and experimental work function values of some compounds. The squared Pearson correlation coefficient of linear fitting excels 0.93.

Actually, previous researches have shown that small changes in stoichiometry can lead to significant changes in the work function (Greiner et al., 2012). As you suggested, we have used DFT to investigate the impact of crystal structure on the calculated work function of some typical minerals (Fig. R1-2). It is found that the standard deviation of work function values of minerals with the same chemical composition but different crystal structure is less than 0.6 eV. However, different chemical compositions can bring about significant changes in work function (Fig. R1-2a), whose relative magnitude can be well predicted by pressure-modulated relative work function proposed in this work (Fig. R1-2b). So, although the crystal structure and surface electronic states do affect the Fermi level position and thus the work function, the constituent atoms and their stoichiometry have a dominant influence. We have incorporated these results and discussions into the revised manuscript.

It should be pointed out that in face of natural complex minerals, there are numerous factors affecting the magnitude of work function (e.g., ambient medium, phase change, deformation, etc.). Therefore, when we consider the law of material change on such a macro scale as the Earth system, it is of little significance to consider more detailed and fine influencing factors (e.g., substitution of trace or minor elements, defects, etc.). Instead, we should focus on evaluating the dominant factors that influence the primary changing trend of mineral work functions. In this respect, the work function estimation method used in this paper is acceptable and appropriate.

Fig. R1-2 (a) The work function of some typical minerals with polymorphism at zero pressure (calculated by DFT). The selected surface corresponds to the most referential crystal plane with the lowest energy. (b) The comparison between the values of pressure-modulated relative work function and the average values of DFT-calculated work function. The squared Pearson correlation coefficient of linear fitting excels 0.97.

The main point of this study is to further extend Dong's electronegativity under four pressure cases (0, 50, 200 and 500 GPa) to those under arbitrary pressure within 500 GPa by the deep learning method. On this basis, a unified formula was firstly regressed to quantitatively describe the relationship between Dong's electronegativity with pressure and electronic configuration. Based on the mathematical formula, therefore, the control rules of electronegativity changes under different pressures can be clearly displayed and physically interpreted in a quantitative and more comprehensible way. Moreover, it greatly simplifies the previous expression of element electronegativity, so that conventional physical quantities such as the number of valence electrons, atomic number, the principal quantum number etc., can be used to estimate the electronegativity under pressure. This method is innovative and will hopefully be extended to other similar fundamental physicochemical properties (e.g., bond energy, covalent radius, polarization, etc.) of elements and their compounds under pressure. Considering the quite important application significance of electronegativity under different pressures in the deep Earth, we presented two case studies related to the element electronegativity (one concerning high and abnormal conductivity phenomena related to deep geophysics, and the other focusing on the prediction and interpretation of electron transfer phenomena by mineral work functions related to deep geochemistry), both of which demonstrated Dong's electronegativity had great application prospect in high-pressure mineralogy. We believe that the importance of this work is not limited to the calculation and application of the two examples in this paper, and more application values should be explored in the future.

References:

- Chen, E. C., Wentworth, W. E., & Ayala, J. A. (1977). The relationship between the Mulliken electronegativities of the elements and the work functions of metals and nonmetals. *The Journal of Chemical Physics*, 67(6), 2642-2647.
- Greiner, M. T., Chai, L., Helander, M. G., Tang, W. M., & Lu, Z. H. (2012). Transition metal oxide work functions: the influence of cation oxidation state and oxygen vacancies. *Advanced Functional Materials*, 22(21), 4557-4568.
- Naseri, M., Bafekry, A., Faraji, M., Hoat, D. M., Fadlallah, M. M., Ghergherehchi, M., ... & Gogova, D. (2021). Two-dimensional buckled tetragonal cadmium chalcogenides including CdS, CdSe, and CdTe monolayers as photo-catalysts for water splitting. *Physical Chemistry Chemical*

Physics, 23(21), 12226-12232.

Nethercot Jr, A. H. (1974). Prediction of Fermi energies and photoelectric thresholds based on electronegativity concepts. *Physical Review Letters*, 33(18), 1088.

Radha, R., Kulangara, R. V., Elaiyappillai, E., Sridevi, J., & Balakumar, S. (2019). Modulation in the band dispersion of Bi₂WO₆ nanocrystals using the electronegativity of transition elements for enhanced visible light Photocatalysis. *Crystal Growth & Design*, 19(11), 6224-6238.

Xu, Y., & Schoonen, M. A. (2000). The absolute energy positions of conduction and valence bands of selected semiconducting minerals. *American mineralogist*, 85(3-4), 543-556.

4. Charge transfer is also an important factor for the work function. The Mulliken electronegativity rely on the charge state of the atoms and ions. The electronegativities are different for Na⁺ and Na, Ca²⁺ and Ca, Mg²⁺ and Mg by the octet rule. The neglects of these facts lead to some misunderstanding conclusion, such as “More importantly, those silicate minerals that have fewer connective silicon-oxygen tetrahedra and larger structure space to accommodate alkali metals alkaline earth metals with lower electronegativity like Ca, Mg, Na.”

Reply: We really appreciate this comment, and agree with you that the electronegativity value relies on the charge state of elements. In general, cations have larger electronegativity values than its neutral atoms, which is attributed to the fact that they become more difficult to further lose electrons; while the opposite is true for anions (Pearson, 1988; Chattaraj and Duley, 2010). The electronegativity used in our dataset is only suitable for neutral atoms rather than cations or anions, as we emphasized in the revised manuscript, e.g., “The predicted results showed that the electronegativity (denoted as χ) of all elements decreased with increasing pressure (some representative elements are shown in Fig. 2a), which is ascribed to the repulsion of nuclei against compression, making it easier to remove electrons from neutral atoms (Lines 143-146 in the revised manuscript)”.

As for cations and anions in compounds, it should be pointed out that their valence state is strongly affected by its bonded counterpart. For example, the valence state of Mg in MgO is +2, but that in MgSe should small than +2, which is attributed that they form ionic and covalent bonds, respectively. Therefore, it is impossible to evaluate the capability of a single ion to gain or lose electrons, instead the compound to which it belongs should be considered. The work function is essentially the parameter of electronic state considering all constituent elements. For example, the predicted work function of Fe, FeO, Fe₃O₄ and Fe₂O₃ at 0 GPa is 1.3, 3.1, 3.3 and 3.4 eV, respectively, corresponding the valence state of Fe is 0, +2, +2.7 and +3, respectively. It clearly shows that the average electronegativity (i.e., work function) increases as the valence state of Fe increases. So, the relative redox state of the constituent ions can be reflected in the work function value.

Inspired by your professional opinion, we added more discussion about the work function after the case study of Fe oxides, as follows: “Considering that work function is a critical parameter related to the average oxidation state of constituent atoms²³, it is feasible and effective to employ $PR-\widehat{W}_F$ as a new metric to evaluate the redox state of a single mineral phase or to predict the tendency and direction of electron transfer between different minerals under the same pressure (Lines 249-253 in the revised manuscript)”. Moreover, the sentence you mentioned has been removed.

References:

Chattaraj, P. K., & Duley, S. (2010). Electron affinity, electronegativity, and electrophilicity of atoms and ions. *Journal of Chemical & Engineering Data*, 55(5), 1882-1886.

Pearson, R. G. (1988). Absolute electronegativity and hardness: application to inorganic

chemistry. *Inorganic chemistry*, 27(4), 734-740.

Hydrogen transfer:

5. Their logic connection of work function and H₂-producing activity is not solid and need more proofs. Perhaps when H₂O meet the minerals, the charge transfer occurs by the work function difference. But the authors should prove the work function difference is big enough and the charge transfer is sufficient for the reduction of H₂O to H₂. If the charge transfer is quite small and fail to produce H₂, their causal relationship between work function difference and reaction to water is not solid as they claimed.

Reply: Yes, our intention is the same as yours. The comparison of electronegativity and work function is used to quantitatively evaluate the electron transfer tendency and direction under the same pressure, and is not a measure of certainty that chemical reactions will occur. What it actually tells us is what kind of mineral is more likely to transfer electrons in the same situation. For example, according to the electronegativity of metal elements, we can know the redox reactivity between metals and H₂O decreases in the following order: K>Ca>Na>Mg>Al>Zn>Fe, which has been proved experimentally.

It should be pointed out that natural oxides and silicates accounts for more than 90% of the total weight of Earth's crust, while silicates made of olivine, pyroxene, hornblende and clays are the dominant rock-forming minerals. So, we have made comparisons of work function between water and some selected silicates and Fe oxides. Moreover, we employed those reported geochemical experiments and real geological records to verify our prediction. Our results show that the work function difference between these Fe(II)-bearing minerals and water will gradually increase with pressure (Fig. R1-3), but with different increasing amplitude. The reactivity of electron transfer from Fe(II)-bearing minerals to H₂O can be determined in the following order: wüstite > magnetite > olivine > pyroxene > hornblende > phyllosilicate > siderite (Fig. R1-3). H₂ could be a product of electron transfer, and the reactivity order of H₂ production by some minerals has been experimentally proved: magnetite > olivine > pyroxene (Mayhew et al., 2013). As you pointed out, combining this experimental evidence with theoretical threshold conditions for H₂ production, a more definite prediction of the possibility of H₂ production or the sequence of H₂ producing activities by these minerals will be given.

In the revised manuscript, we emphasized that the work function difference can be applied to quantitatively comparing the tendency and directions of electron transfer between different minerals. Before confirmed by experimental data, this rule applies primarily to judging reactants (e.g. minerals and H₂O) rather than reaction products (e.g., H₂). In the revised manuscript, we put more emphasis on the electron transfer tendency and directions between minerals and H₂O, while H₂ is less mentioned and just treated as a potential product. For example, "This well interpreted the deep high-conductivity anomalies, and enabled us to discover the universal H₂ production potential of OH-bearing silicates during ongoing subduction" in *Abstract* has been corrected as "This well explains the deep high-conductivity anomalies, and enables us to discover the redox reactivity between widespread Fe(II)-bearing minerals and water during ongoing subduction (Lines 33-35 in the revised manuscript)".

References:

Mayhew, L. E., Ellison, E. T., McCollom, T. M., Trainor, T. P., & Templeton, A. S. (2013). Hydrogen generation from low-temperature water-rock reactions. *Nature Geoscience*, 6(6), 478-484.

Fig. R1-3 The pressure-modulated relative work function ($\text{PR-}\widehat{W}_F$) between Fe(II)-bearing minerals and water as the function of pressure. The composition of silicate minerals refers to International Mineralogical Association (IMA) database (<https://rruff.info/ima/>). The symbol “Wus”, “Mt”, “Ol”, “Px”, “Hbl”, “PLS”, “Sd” represents wüstite, magnetite, olivine, pyroxene, hornblende, phyllosilicate, siderite, respectively.

6. Following by comment #4, based on HSAB (hard soft acid base) theory, I consider the theory based on chemical hardness is more reliable, because most mineral reactions with H_2O have no charge transfer.

Reply: We really appreciate your recommended HSAB (hard soft acid base) theory. Actually, what we are focusing on is electron transfer from one element or compound to the other, so all reactions in this work are redox-related electron transfer (please see the following equations used as case studies in this work). In the revised manuscript, we focused on redox reactions between Fe(II)-bearing minerals and water as case studies. Considering that electronegativity is based on the electron configuration of atoms and can directly indicate the distribution of electron density, redox potential and reactivity of elements, it will be more dependable and intuitive to use electronegativity as a proxy to predict and interpret redox reactions.

Moreover, it should be pointed out that solid-state analogs of Mulliken electronegativity and chemical hardness are the work function and the band gap, respectively (Dong et al., 2022). As for redox reactions involved by solid minerals, electronegativity-based work function might be a more suitable proxy for prediction. Many thanks for your reminding, and the chemical hardness of elements deserves further investigation in our future work. In this revised manuscript, we have added the application restrictions of electronegativity and work function when predicting electron transfer-related redox reactions.

References:

- Dong, X., Oganov, A. R., Cui, H., Zhou, X. F., & Wang, H. T. (2022). Electronegativity and chemical hardness of elements under pressure. *Proceedings of the National Academy of Sciences*, 119(10), e2117416119.
- Seyfried Jr, W. E., Foustoukos, D. I., & Fu, Q. (2007). Redox evolution and mass transfer during serpentinization: An experimental and theoretical study at 200 C, 500 bar with implications for ultramafic-hosted hydrothermal systems at Mid-Ocean Ridges. *Geochimica et Cosmochimica*

Acta, 71(15), 3872-3886.

Zema, M., Ventruti, G., Lacalamita, M., & Scordari, F. (2010). Kinetics of Fe-oxidation/deprotonation process in Fe-rich phlogopite under isothermal conditions. *American Mineralogist*, 95(10), 1458-1466.

7. The authors claimed, “Such predictions are in good agreement with experimental observations that the metallic Fe, FeO, Fe₂O₃ and FeO₂ begin to react with H₂O at 5, 78, 110 GPa and over 130 GPa, respectively”. But by the reaction equations supplied in Fig 4a, some of the equations (Fe+H₂O→FeO+FeH_x) has charge transfer and some have none (Fe₂O₃+H₂O→FeO₂H_x, in fact only x=1 can be balanced). The reaction mechanisms are different. Their prediction of the transfer of electrons from FeO_x to H₂O cannot explain these all.

Reply: Thank you so much for this comment. We are sorry for providing this confusing equation (Fe₂O₃+H₂O→FeO₂H_x) that lacks a clear explanation. In fact, it is still a redox reaction. The valence state of Fe and O in the peroxide product FeO₂H_x is +2 and -1, respectively, after the electron transfer between Fe₂O₃ and H₂O (Liu et al., 2019). The H atom keeps a valence state of +1 and does not bond with Fe or O under high pressures (e.g., 78 GPa), but could be itinerant in the peroxide crystal lattice and move freely like a fluid within an oxygen framework to reaching a superionic state (Hou et al., 2021). However, as you reminded, the reaction mechanisms involving intricate redox reactions between water and metallic Fe as well as Fe oxides are different. For example, metallic Fe donates its electron to H₂O, while H₂O and O²⁻ in Fe₂O₃ would instead reduce Fe³⁺ in Fe₂O₃ to Fe²⁺ (Hu et al., 2016). Therefore, after careful consideration, we deleted the contents of electron transfer between FeO_x (with multiple valence states) and H₂O, and focused on the redox reactions between Fe(II)-bearing minerals and water as case studies (Fig. R1-3).

Actually, as we emphasized in the revised manuscript: “Considering that work function is a critical parameter related to the average oxidation state of constituent atoms²³, it is feasible and effective to employ PR- \widehat{W}_F as a new metric to evaluate the redox state of a single mineral phase or to predict the tendency and direction of electron transfer between different minerals under the same pressure (Lines 248-252 in the revised manuscript)”, PR- \widehat{W}_F is suitable for two reactants with sole electron transfer pathway, while multiple electron transfer pathways (e.g. Fe₂O₃ and H₂O in the above case) are too complicated to be resolved only by PR- \widehat{W}_F .

References:

- Hou, M., He, Y., Jang, B. G., Sun, S., Zhuang, Y., Deng, L., ... & Mao, H. K. (2021). Superionic iron oxide–hydroxide in Earth’s deep mantle. *Nature Geoscience*, 14(3), 174-178.
- Hu, Q., Kim, D. Y., Yang, W., Yang, L., Meng, Y., Zhang, L., & Mao, H. K. (2016). FeO₂ and FeOOH under deep lower-mantle conditions and Earth’s oxygen–hydrogen cycles. *Nature*, 534(7606), 241-244.
- Liu, J., Hu, Q., Bi, W., Yang, L., Xiao, Y., Chow, P., ... & Mao, W. L. (2019). Altered chemistry of oxygen and iron under deep Earth conditions. *Nature communications*, 10(1), 1-8.

8. Work functions in Fig4b cannot give the results about H₂ production, also cannot be a scale of acidity. OH is an unstable free radical, not a stable compound. The differences between OH and minerals lack of exact physical meanings.

Reply: Yes, we agree with you. There are some inaccurate statements here. As replied in comment #5, these curves in Fig. R1-3 actually indicate pressure-enhanced reactivity of Fe(II)-bearing minerals with H₂O and significant individual difference. In the revised manuscript, we focused on discussing the tendency and directions of electron transfer between minerals and H₂O, while H₂ was just regarded as a potential product. The confused scale of “acidity” in the previous

version has been deleted.

As for the issue of OH group, we have corrected it as H₂O molecule in our calculations, as displayed in Fig. R1-3, which actually indicates the reactivity between minerals and surrounding water. However, in most of hydrous minerals (e.g., mica and hornblende) and even at dozens of gigapascals, OH groups are directly bonded to metals such as Fe, Mg and Ca (Libowitzky and Beran, 2006). These OH groups are termed as “constitutional water” in mineralogy, which should be the most direct acceptors of valence electrons from bonded metals. From this point of view, the metal-to-ligand (OH) electron transfer also deserves our attention, which might bring about dehydrogenation, following H₂ production and even decomposition of hydrous silicate minerals (please see the flowing equation from Zema et al., 2010; Lempart et al., 2018). So, in this revised manuscript, we have reorganized this session and divided it into two levels according to electron donors and acceptors: (1) minerals and their surrounding H₂O molecules, (2) metals and their OH ligands. Please see detailed revisions in the session “*Work function-informed redox reactivity between Fe(II)-bearing minerals and water under high pressure*” of the revised manuscript (Lines 328-407 in the revised manuscript).

Fig. R1-3 The pressure-modulated relative work function ($PR-\widehat{W}_F$) between Fe(II)-bearing minerals and water as the function of pressure. The composition of silicate minerals refers to International Mineralogical Association (IMA) database (<https://rruff.info/ima/>). The symbol “Wus”, “Mt”, “Ol”, “Px”, “Hbl”, “PLS”, “Sd” represents wüstite, magnetite, olivine, pyroxene, hornblende, phyllosilicate, siderite, respectively.

References:

- Lempart, M., Derkowski, A., Luberd-Durnaś, K., Skiba, M., & Błachowski, A. (2018). Dehydrogenation and dehydroxylation as drivers of the thermal decomposition of Fe-chlorites. *American Mineralogist: Journal of Earth and Planetary Materials*, 103(11), 1837-1850.
- Libowitzky, E., & Beran, A. (2006). The structure of hydrous species in nominally anhydrous minerals: Information from polarized IR spectroscopy. *Reviews in Mineralogy and Geochemistry*, 62(1), 29-52.
- Zema, M., Ventrucci, G., Lacalamita, M., & Scordari, F. (2010). Kinetics of Fe-oxidation/deprotonation process in Fe-rich phlogopite under isothermal conditions. *American Mineralogist*, 95(10), 1458-1466.

Reviewer #2 (Remarks to the Author):

Comments:

The authors developed a Dense Attention Network (DAN) framework to predict the electronegativity of 96 elements under high pressure conditions. With the predicted electronegativity, the work function of 5828 minerals were further evaluated and were correlate with the electron transfer mechanism of minerals in deep Earth's interior. The introduction of the DAN algorithms was well-written, and the authors demonstrated systematic and robust methods to prevent overfitting of the DAN model. However, the description of the establishment and the validation of dataset is ambiguous. If a data set could not be well verified, the data-driven results would be unreliable. With this in mind, I believe that the following points should be addressed prior to publication to enhance the impact of the manuscripts.

Reply: Many thanks for your recognition of our work and your professional and constructive comments. In this revised manuscript, we added more descriptions of the electronegativity dataset and compared it with other definitions of electronegativity to verify its validation. Moreover, in order not to be overstated, we have added some restrictions to the use of the calculated work function. Thanks to all your helpful suggestions, this version has been further improved. All revisions are highlighted in the text, and a point-by-point response is given below.

Detailed comments:

1. A summary about the calculation of electronegativity via the Density Functional Theory (DFT) methods should be provided, instead of just referring to a citation (Ref. 17). Also, since there are several types of electronegativity, the authors should specify that the electronegativity in this work is the Mulliken electronegativity, and briefly comment on the difference between the Mulliken electronegativity and the well-known Pauling electronegativity.

Reply: We really appreciate this suggestion, which greatly contributes to a better understanding of the proposed electronegativity. We have added the summary about the calculation of electronegativity based on the density function theory (DFT) methods in the main text before our results. Because the definition of Mulliken electronegativity (i.e., taking vacuum as the reference) is no longer applicable to the case of high pressure, the newly-calculated electronegativity made a proper modification on the Mulliken electronegativity. However, it is still linearly correlated with the well-known Pauling electronegativity (please see the following Fig. R2-1), and is proved to work well under high pressures. In the revised manuscript, we have added more descriptions of the differences between the electronegativity dataset we used and other types of electronegativity. For your convenience, we copied the revised text below for you further review (Lines 95-109 in the revised manuscript).

“The newly-calculated electronegativity from Dong and his co-workers appropriately modified the definition of Mulliken electronegativity for better use at high pressures, and it used the enthalpy as the relevant thermodynamic potential within the framework of homogeneous electron gas model. At zero pressure, Dong's electronegativity has the same energy unit as Mulliken electronegativity (eV) and also has a good linear correlation with Pauling electronegativity (correlation coefficient $R = 0.91$). For example, the electronegativity of rubidium (Rb), zinc (Zn), nitrogen (N) in Pauling scale is 0.8, 1.6 and 3.0, respectively; while that is -0.8 , 1.0 and 4.4 eV, respectively in the Dong's definition. Although Dong's electronegativity dataset has different reference systems and specific data from the other two well-known electronegativity datasets, it successfully extends the applicability of evaluating electron transfer trends between elements to high pressure cases. The predicted stability of Mg_xFe_y compounds at pressures above 100 GPa accords well with the experimental observations¹⁷, indicating Dong's electronegativity after modifying Mulliken

electronegativity is specifically suitable for high pressure cases”.

Fig. R2-1 Correlation between the calculated electronegativity and Pauling electronegativity. Solid lines show linear fits: $y = 2.29x - 2.88$ (correlation coefficient $R = 0.91$) (Dong et al., 2022).

References:

Dong, X., Oganov, A. R., Cui, H., Zhou, X. F., & Wang, H. T. (2022). Electronegativity and chemical hardness of elements under pressure. *Proceedings of the National Academy of Sciences*, *119*(10), e2117416119.

2. The reported values of electronegativity are not consistent with the common sense about electronegativity. For instance, the electronegativity of fluorine in Extended data Fig. 2 is around seven, but the well-known value should be around four. This deviation may be attributed to the different definitions of electronegativity or the error from DFT calculations. If the author can clarify the deviation, it will avoid the misunderstanding by readers.

Reply: Yes, you're absolutely right that our used electronegativity values are different from the well-known Pauling electronegativity. For example, the electronegativity of rubidium (Rb), zinc (Zn), nitrogen (N) in Pauling scale is 0.8, 1.6 and 3.0, respectively, while that in our dataset is -0.8 , 1.0 and 4.4 eV, respectively. As replied above, the concept of Pauling electronegativity is not suitable for high-pressure cases, so a modified electronegativity with a different reference system from Pauling electronegativity was put forward by Dong et al. (2022). Although the values of the two reference systems differ at zero pressure, they are definitely comparable (Fig. R2-1). For example, the electronegativity of O and S in Pauling's scale is 3.5 and 2.5, respectively; that in Mulliken's scale is 9.6 and 7.4 eV, respectively; while that in our used scale is 4.8 and 3.2 eV, respectively. Whatever definition of electronegativity is used, oxygen has more positive electronegativity than S, indicating O has stronger ability of accepting electrons in comparison with S. So, the absolute electronegativity value of an element alone is of little significance, while the relative electronegativity of two atoms could be applied to quantitatively evaluating the tendency and directions of electron transfer between them. In general, the comparison of the relative magnitude of electronegativity would not be affected by the choice of reference system. According to your comments, we added more introduction on how the electronegativity we used compared to Pauling electronegativity (please see details and specific revision in the above reply).

References:

Dong, X., Oganov, A. R., Cui, H., Zhou, X. F., & Wang, H. T. (2022). Electronegativity and chemical hardness of elements under pressure. *Proceedings of the National Academy of Sciences*, *119*(10), e2117416119.

3. In Fig. 2a, the machine learning (ML) predicted electronegativity considerably decreases in the pressure regime from 0 GPa to 20 GPa. However, the ML data set only includes the electronegativity at 0 GPa and 50 GPa. The authors should verify that the sharp reduction of the electronegativity is not the artefact due to the lack of training data in this pressure regime.

Reply: It's a great and quite professional suggestion. Our training dataset is based on the electronegativity values at 0, 50, 200 and 500 GPa. According to the limited data in the reference (Dong et al., 2022) (Fig. R2-2), we noted that the electronegativity of some elements sharply reduced with pressure in the range of 0-20 GPa, as you pointed out, while some like Li and Na considerably decreased from 0 to 100 GPa, similar to our prediction. Considering that these 7 representative elements belong to s-, p-, ds- and d-block of the periodic table, we can use them as a small dataset of truth values to make a comparison with our predicted values through deep learning. We found that our predicted electronegativity values are highly consistent with those calculated by the DFT even though our deep learning model was not based on these 35 data points. In other words, the sharp reduction of our predicted electronegativity is not the artefact, indicating our model has a brilliant generalization ability. That's really what deep learning can do in this case, i.e., it can obtain reliable data before time-consuming DFT calculations.

Further, in order to ensure the generalization of the model and the reliability of the conclusions on the small sample data set, we used several methods to ensure the generalization performance of the model, including feature augmentation methods (operation augmentation and symbolic transformer), model mechanisms (self-attention and dense connection) and ensemble learning method (5-fold bagging). Finally, high performance (RMSE=0.425, MAE=0.283 and $R^2=0.987$) was achieved in the testing set.

Thank you for pointing out that, we have added new comparison with these 35 data points (as discussed above) in the revised manuscript. For your convenience, we copied them as follows: "Besides, the predicted electronegativity values were also compared with those DFT-calculated electronegativity of Li, C, N, Na, Mg, Ni and Au at 20, 40, 60, 80 and 100 GPa (Extended Data Fig. 2). Although the dataset used in deep learning model was not from these pressure points, the high prediction ability (RMSE=0.425, MAE=0.283 and $R^2=0.987$) demonstrated the deep learning model had an excellent generalization performance (Lines 129-134 in the revised manuscript)".

Fig. R2-2 The calculated electronegativity as a function of pressure (Dong et al., 2022).

Fig. R2-3 Performances of deep learning model on the testing set consisting of Na, Mg, Ni, C, N, Li and Au at 20, 40, 60, 80 and 100 GPa.

References:

Dong, X., Oganov, A. R., Cui, H., Zhou, X. F., & Wang, H. T. (2022). Electronegativity and chemical hardness of elements under pressure. *Proceedings of the National Academy of Sciences*, 119(10), e2117416119.

4. The evaluation of work function based on the arithmetic mean of the electronegativity values of constituent atoms may be a reasonable strategy for pure mental elements, but it could be an oversimplified approach for the minerals with hydrogen and oxygen atoms, which are sensitive to their morphology or their molecular environments. If the authors can provide a comparison between their method and the other robust approaches for the calculation of work of some common minerals, the results and discussion will be more convincing.

Reply: Many thanks for this comment and suggestion. Actually, as we mentioned in the manuscript, one of the most classical ways to estimate the work function of a compound is to calculate the geometric mean of the absolute electronegativity values of its constituent atoms when the chemical formula is given (Nethercot, 1974; Chen et al., 1977; Xu and Schoonen, 2000). The work function values estimated by this method accord well with the experimental values of dozens of compounds which is made of different elements (please see table R2-1 from Chen et al., 1977). However, the electronegativity values of elements are generally negative under high pressure, so the work function calculation method based on geometric mean value is no longer applicable. In this work, we have moderately modified the work function at high pressure by using the arithmetic mean of the electronegativity values of constituent atoms. Actually, the arithmetic mean is another indicator to well reflect the mean electronegativity level of the constituent atoms, which is just slightly higher than or equal to the arithmetic mean. We calculated the relative work functions of some compounds for which the experimental values were available in the literatures, and found that they were approximately linearly related with the experimental values (Fig. R2-4). So, the calculated work function in this work can reflect the relative magnitudes, and is suitable for predicting the tendency and direction of electron transfer between different compounds under the same pressure.

Table R2-1 Geometric mean of the electronegativities (M_A and M_B) (i.e., estimated work function) and solid state data for elements and compounds. The difference between the photoelectric threshold (E_i) and the half of band gap (E_g) is the experimental work function. The difference between estimated work function and experimental one is defined as $\delta = E_i - 1/2E_g - (M_A M_B)^{1/2}$ (Chen et al., 1977).

Element or compound	$(M_A M_B)^{1/2}$ (eV)	$E_t - \frac{1}{2}E_f$ (eV)	δ (eV)
Si $s^2 p^2 V_2, p$	4.20	4.55	0.35
Ge $s^2 p^2 V_2, p$	3.95	4.46	0.51
KF	5.00	4.94	-0.06
NaCl	4.81	4.19	-0.62
KCl	4.46	4.36	-0.10
CsCl	4.21	3.46	-0.75
KBr	4.27	4.47	0.20
CsBr	4.04	3.34	-0.70
LiI	4.54	4.42	-0.12
NaI	4.35	4.41	0.06
KI	4.03	4.19	0.16
RbI	3.96	4.12	0.16
CsI	3.81	3.35	-0.46
InP	5.41	5.28	-0.13
GaAs	5.37	5.05	-0.32
InAs	5.34	5.37	0.04
AlSb	4.90	4.98	0.08
GaSb	4.97	4.95	-0.02
InSb	4.95	5.28	0.33
ZnO	6.53	6.37	-0.16
ZnS	5.74	5.78	0.04
ZnSe	5.58	5.50	-0.08
ZnTe	5.39	4.89	-0.50
CdS	5.67	6.05	0.38
CdSe	5.51	5.85	0.34
CdTe	5.32	5.22	-0.10
HgS	6.02	6.00	-0.02
HgSe	5.85	5.61	-0.24
HgTe	5.65	5.97	0.32

Fig. R2-4 Work function calculation of some compounds for the comparison of their experimental data. The squared Pearson correlation coefficient exceeds 0.93 of linear fitting.

Regarding to the morphology and molecular environments mentioned by you, we have used DFT method to investigate the impact of crystal structure, exposed crystal planes and surface adsorbed species on the calculated work function of some common compounds (Fig. R2-5 and Fig. R2-6). It is found that the standard deviation of work function values of minerals with the same chemical composition but different crystal structure is less than 0.6 eV. However, different chemical composition can bring about significant changes in work function (Fig. R2-5a), whose relative magnitude can be well predicted by the pressure-modulated relative work function proposed in this work (Fig. R2-5b). So, although the crystal structure and surface electronic states do affect the Fermi level position and thus the work function, the constituent atoms and their stoichiometry have a dominant influence.

However, based on the DFT calculations, the surface bonded with protons (i.e., forming OH group for oxides) on mineral surface, which is one of cases of changed molecular environments, can greatly impact on work function values (Fig. R2-6). The considerable decreasing of work function is not ascribed to the changing chemical composition when incorporated with protons

because the nonmetal hydrogen would generally give rise to the increasing work function. The reason could be the electronic interaction between metal-bearing mineral with surface OH groups, which greatly affects surface electronic state. Therefore, unlike the above-mentioned morphology factor, different molecular environments especially those altering electronic state would result in various degrees of changing work function.

It should be pointed out that in face of natural complex minerals, there are numerous factors affecting the magnitude of work function (e.g., ambient medium, phase change, deformation, etc.). Therefore, when we consider the law of material change on such a macro scale as the Earth system, it is of little significance to consider more detailed and fine influencing factors (e.g., substitution of trace or minor elements, defects, etc.). Instead, we should focus on evaluating the dominant factors that influence the primary changing trend of mineral work functions. In this respect, the work function estimation method used in this paper is acceptable and appropriate. We believe that the importance of this work is not limited to the calculation and application of work function, there should be more potential application to be explored in the future.

Fig. R1-2 (a) The work function of some typical minerals with polymorphism at zero pressure (calculated by DFT). The selected surface corresponds to the most referential crystal plane with the lowest energy. (b) The comparison between the values of pressure-modulated relative work function and the average values of DFT-calculated work function. The squared Pearson correlation coefficient of linear fitting excels 0.97.

Fig. R2-6 DFT-based calculation of work function for brookite and anatase minerals with clear surface or surface saturated with H^+ species.

After your professional comments and suggestions, we have made corresponding revision in this revised manuscript. To distinguish our calculated relative work function from the absolute or

experimental one, we named it as pressure-modulated relative work function (denoted as $\text{PR-}\widehat{W}_F$) in the revised manuscript. Moreover, in the context involving the definition of relative work function and discussion of new DFT-calculation results, we have added strict restrictions to avoid misunderstanding or misuse. The specific revisions are as follows:

“This estimation actually yields the relative work function, which showed approximate linearity with the experimental values (Extended Data Fig. 4). Since $\text{PR-}\widehat{W}_F$ is derived from element electronegativity, it also depends on the selected reference system. In practice, we should use its relative value for comparison rather than the absolute value, because the real work function is extremely hard to measure (Lines 243-248 in the revised manuscript)”.

“Actually, there are many structural factors besides chemical composition that can alter the specific value of the work function. To this end, density functional theory (DFT) was used to investigate the impact of crystal structure and exposed surfaces on the calculated work function (Extended Data Fig. 5). It is found that work function values of different minerals with the same chemical composition but different crystal structure or exposed surfaces are still similar, with a standard deviation under 0.6 eV. Moreover, different chemical composition can cause significant work function changes, and we can predict its relative magnitude well by using $\text{PR-}\widehat{W}_F$. Therefore, even though crystal structure and surface electronic state do affect the Fermi level position and thus the work function, the constituent atoms and their stoichiometry have a dominant influence when we compare different compounds and reveal general rules (Lines 253-263 in the revised manuscript)”.

“Based on DFT calculations, it is found that oxides and silicates minerals that are superficially bonded with protons (i.e., forming adsorbed OH groups) would have considerably decreasing work function by more than 2 eV (Extended Data Fig. 6). Such a change in work function actually comes from the change in the surface molecular environment and electronic state, suggesting the robust electronic interaction between metal-bearing minerals and surface OH groups (Lines 353-359 in the revised manuscript)”.

References:

- Chen, E. C., Wentworth, W. E., & Ayala, J. A. (1977). The relationship between the Mulliken electronegativities of the elements and the work functions of metals and nonmetals. *The Journal of Chemical Physics*, 67(6), 2642-2647.
- Nethercot Jr, A. H. (1974). Prediction of Fermi energies and photoelectric thresholds based on electronegativity concepts. *Physical Review Letters*, 33(18), 1088.
- Xu, Y., & Schoonen, M. A. (2000). The absolute energy positions of conduction and valence bands of selected semiconducting minerals. *American mineralogist*, 85(3-4), 543-556.

5. At page 9, the authors said that “Consistent with Marcus’ electron transfer theory, the electron transfer tendency in the metal-ligand bonding system strongly depends on the distance between the electron donor and acceptor.” According to Marcus’ theory, electron transfer rate is proportional to the square of electronic coupling, which is dependent on not only distance between the electron donor and acceptor but also their orientation. Under high pressure, as all the atoms are in close contact, the atomic or molecular alignments play a decisive role in electronic coupling that determines the electron transport rate, especially for minerals with non-metal elements. Therefore, the current description at page 9 could be misleading, and it needs to be modified.

Reply: Thank you for your professional advice. As you mentioned, the electronic coupling in Marcus’ theory is mainly controlled by the overlap of electronic wavefunctions, mainly including two factors, one of which is the spatial mutual orientation of donor and acceptor orbitals, and the other is the distance between donor and acceptor (Stuchebrukhov and Marcus, 1995; Taylor and

Kassal, 2018). In general, the electronic coupling between donor and acceptor (i.e., corresponding reaction rates) should decrease approximately exponentially with their distance. By contrast, the orientational dependence of electronic couplings depends on the shape of the orbitals, which varies from molecule to molecule (Taylor and Kassal, 2018). It is easy to link pressure with distance, because that pressure can induce close contact and thus shorten the distance between the donor and acceptor. However, we can't find any previous researches about the effect of pressure on the orbital orientation between donor and acceptor, and it is also difficult to predict a general rule because different elements have different electron orbitals. Even though, we can still say that pressure can at least enhance the overlap of electronic wavefunctions through shortening the distance between the donor and acceptor, thus promoting the reaction rate.

In the revised manuscript, we have added more descriptions about the electronic coupling factor in Marcus' theory. We copied the revisions as follows for your convenience: "Consistent with Marcus' electron transfer theory², the reaction rate of electron transfer between the electron donor and acceptor is proportional to the square of electronic coupling, which strongly depends on the two geometric factors affecting the overlap of electronic wavefunction, i.e., their distance and orbital orientation. Although the orientational dependence on pressure is unknown, it is easy to conclude that pressure could promote reaction rates at least from the perspective of enhancing the overlap of electronic wavefunction through shortening the distance between donor and acceptor (Lines 367-373 in the revised manuscript)".

References:

- Stuchebrukhov, A. A., & Marcus, R. A. (1995). Theoretical study of electron transfer in ferrocycchromes. *The Journal of Physical Chemistry*, 99(19), 7581-7590.
- Taylor, N. B., & Kassal, I. (2018). Generalised Marcus theory for multi-molecular delocalised charge transfer. *Chemical science*, 9(11), 2942-2951.

6. The values of electronegativity and work function are also affected by the morphology or the molecular environment of minerals. In the deep regions of Earth's interior, high temperature can lead to dramatic change of the molecular environment. In this work, it seems that both electronegativity and work function are obtained from the prediction (or evaluation) of individual molecular configuration. The authors should briefly discuss about the impact of temperature and molecular environment.

Reply: Yes, there are complex influencing factors on electronegativity and work function in deep Earth. As replied in comment #4, the roles of morphology and molecular environment in regulating the magnitude of work function were further studied and compared with our proposed pressure-modulated relative work function based on DFT calculations (Fig. R2-5 and R2-6). The results show the morphology factor has a negligible effect, while the changes of the molecular environments, especially the electronic state, will lead to work function change in different degrees. Actually, in face of natural complex minerals, there are numerous factors affecting the magnitude of work function (e.g., ambient medium, phase change, deformation, etc.). Therefore, when we consider the law of material change on such a macro scale as the Earth system, it is of little significance to consider more detailed and fine influencing factors (e.g., substitution of trace or minor elements, defects, etc.). Instead, we should focus on evaluating the dominant factors that influence the primary changing trend of mineral work functions. In this respect, the work function estimation method used in this paper is acceptable and appropriate.

As for the issue of temperature, you are completely right that it can cause dramatic changes in the molecular environments, e.g., disturbing adsorption equilibrium and affecting adsorbing capacity. To this end, we have used DFT calculations and found that the surface saturated with

protons (i.e., forming OH group for oxides) on mineral surfaces, just as one case of changed molecular environments, can greatly influence the work function values and suggest the robust electronic interaction between metal-bearing mineral with surface OH groups (Fig. R2-6). However, when the concentration of surface protons reduced by half, which can be regarded as the situation with different adsorption equilibrium under different temperature, the calculated work function keeps almost the same as the full-saturated case (Fig. R2-6). Therefore, though the temperature affects the molecular environments of minerals, it may play less roles in altering the work function.

Actually, we emphasize here that the main point of this study is the electronegativity of atoms and work function of minerals could be applied to quantitatively evaluating the electron transfer tendency and directions under the same condition (i.e., same pressure or same temperature). In this situation, the comparison of electronegativity and work function is reasonable and has clear meanings.

We have added the DFT calculation results of molecular environment and discussion about the factor of temperature in the revised manuscript, which is copied here for your convenience:

“Based on DFT calculations, it is found that oxides and silicates minerals that are superficially bonded with protons (i.e., forming adsorbed OH groups) would have considerably decreasing work function by more than 2 eV (Extended Data Fig. 6). Such a change in work function actually comes from the change in the surface molecular environment and electronic state, suggesting the robust electronic interaction between metal-bearing minerals and surface OH groups (Lines 353-359 in the revised manuscript)”.

“It should be noted that temperature can not only greatly alter the concentration and state of surface adsorbed species, but also effectively promote the reaction rate according to Arrhenius equation. Further work related to temperature effect on electronegativity and work function deserves more theoretical and experimental efforts. But considering the relativity of electronegativity and work function, the comparison between different minerals under the same conditions (i.e., same pressure or temperature) is of significant practical value (Lines 394-400 in the revised manuscript)”.

Fig. R2-6 DFT-based calculation of work function for brookite and anatase minerals with clear surface or surface saturated with H⁺ species.

REVIEWERS' COMMENTS

Reviewer #1 (Remarks to the Author):

The authors seems to address all of my concerns and make great progress compared with their first edition. By their extension of high pressure electronegativity in theory, I considered it can attract board interest to the readers of Nat. Commun. and suggested to publish in Nat. Commun.

Reviewer #2 (Remarks to the Author):

Comments:

The author has addressed all the reviewers' comments and has clarified the ambiguity we concerned. This manuscript demonstrates a novel machine learning approach that predicts electronegativity and work function under high pressure conditions and provides an opportunity to reveal the electron transfer mechanism of minerals in deep Earth's interior. Therefore, I recommend publication of the manuscript.